



# A glacial lake outburst flood hazard assessment for the Phochhu River Basin, Bhutan

Tandin Wangchuk[1,2], Ryota Tsubaki[2]

[1]National Center for Hydrology and Meteorology, Bhutan

[2]Department of Civil and Environmental Engineering, Nagoya University, Japan

*Correspondence to*: Ryota Tsubaki (rtsubaki@civil.nagoya-u.ac.jp)

**Abstract.** The melting of glaciers has led to an unprecedented increase in the number and size of glacial lakes, particularly in the Himalayan region. A Glacial Lake Outburst Flood (GLOF) is a natural hazard in which water from a glacial or glacier-fed lake is swiftly discharged. GLOFs can significantly harm life, infrastructure, and settlements located downstream, and can 10 cause considerable ecological, economic, and social impacts. Based on a dam breach model, BREACH, and a hydrodynamic model, HEC-RAS, we examined the potential consequences of a GLOF originating from the Thorthomi glacial lake, located within the Phochhu River Basin, one of Bhutan's largest and rapidly expanding glacial lakes. Our analysis revealed that, following a breach, the Thorthomi glacial lake will likely generate a peak flow of 16,360 $m^3\,s^{-1}$ within four hours. Such discharge could potentially cause considerable damage, with an estimated 245 hectares of agricultural land and over 1,277 15 buildings at risk for inundation. Our results emphasize an urgent need for understanding and preparing for the potential consequences of a GLOF from Thorthomi Lake in order to mitigate ecological, economic, and social impacts on downstream areas. Our findings provide valuable insights for policymakers and stakeholders involved in disaster management and preparedness.

## 1 Introduction

### 1.1 Background

Floods are one of the most common natural disasters worldwide and can cause extensive socio-economic damage. Globally, floods affected approximately 2.3 billion people and cause an estimated 622 billion (USD) in damage in the last two decades (UNISDR, 2015). Glacial Lake Outburst Floods (GLOFs) are floods caused by a sudden water release from glacial or glacier-fed lakes and cause a rapid rise in water level within a short time in downstream areas, resulting in devastating consequences 25 (Gurung et al., 2017; Komori et al., 2012). GLOFs are infrequent but highly destructive natural disasters that are difficult to predict. Prior to their occurrence, their damage is also difficult to predict. The recent acceleration of glacier melting and recession, primarily driven by climate change, has led to a significant increase in the number of moraine-dammed (natural dams formed by glacial processes) glacial lakes over the past few decades (Sattar et al., 2021; Westoby et al., 2014; Worni et al., 2014). In particular, due to climate warming (Gardelle et al., 2011), the Eastern Himalayan area has seen a significant





increase in the number and area of glacial lakes, increasing the vulnerability of nearby communities to potential GLOF impacts within this region (Hagg et al., 2021). Taylor et al. (2023) estimated that approximately 15 million people are exposed to risks associated with potential GLOFs and that most of these populations are concentrated within High Mountain Asian (HMA) areas.

Based on the latest report from the National Center for Hydrology and Meteorology (NCHM) in Bhutan, 567 glacial lakes in

the country span an area of 55.04 km$^2$, accounting for 19.03% of the total number of water bodies (NCHM, 2021). In 2001, the Department of Geology and Mines (DGM) in Bhutan and the International Center for Integrated Mountain Development (ICIMOD) performed the first-ever inventory of glaciers, glacial lakes, and Potentially Dangerous Glacial Lakes (PDGLs); and identified 24 glacial lakes as PDGLs (Mool et al., 2001). However, in 2019, NCHM reassessed the number of PDGLs using field-verified data and the latest high-resolution Sentinel 2 satellite images and revised the number to 25, of which eight

are considered safe based on lake morphology, surrounding features, bathymetry conditions, and associated feeding glaciers (NCHM, 2019b). Figure 1 displays rivers and the river basin system in the area, together with the distribution of glaciers and glacial lakes in Bhutan. The Punatsangchhu River Basin contains eleven PDGLs (the largest in the country). The Phochhu sub-basin contains nine PDGLs, making it a hotspot for GLOFs and glacial related disasters.

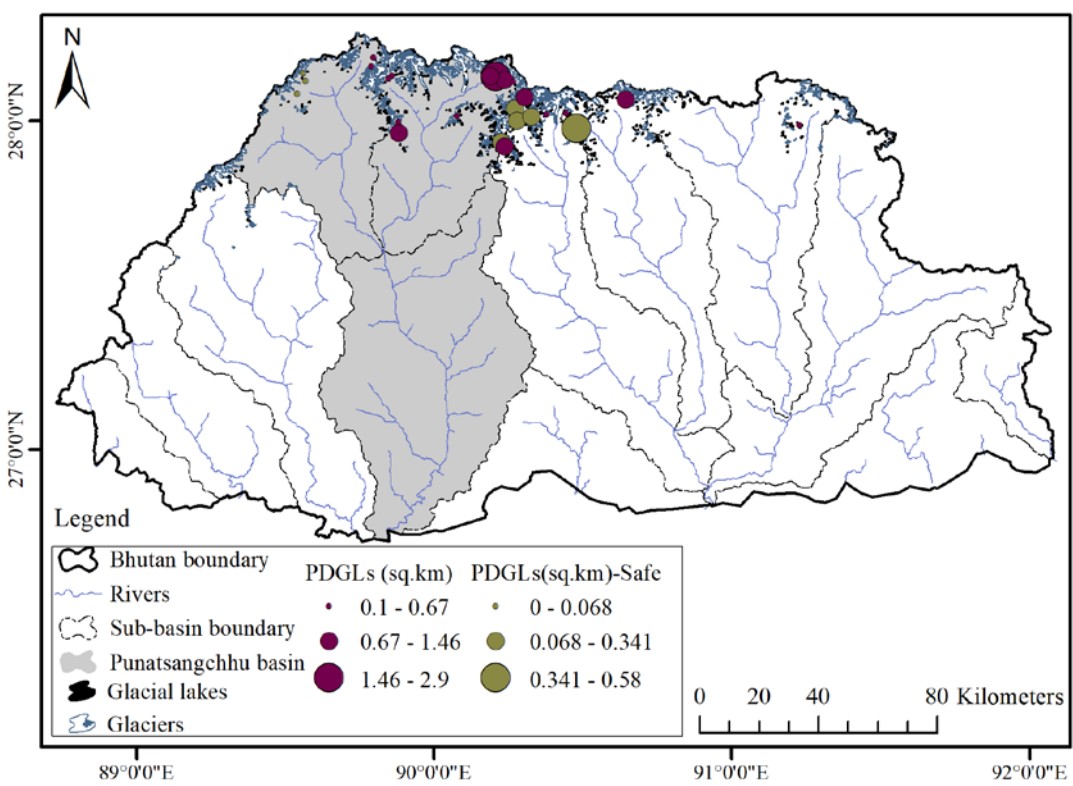


**Figure 1.** A map showing rivers and the basin system of Bhutan, as well as the distribution of potentially dangerous glacial lakes. Bubbles are scaled to the total lake area and color-coded for classification. Data source: National Center for Hydrology and Meteorology (NCHM).





Bhutan has faced several GLOF events in the past. However, many of these events were either not reported or documented. One of the most catastrophic GLOFs took place on 6 October 1994, when the moraine dam of Luggye Lake partially collapsed,

leading to the release of a massive amount of water and debris downstream, destruction to infrastructure and farmland, and the deaths of 21 people (Watanabe and Rothacher, 1996; Leber et al. 2000). Another significant GLOF incident occurred in 2009, when an outburst from Tshojo Lake, located at the headwaters of the Phochhu River, caused downstream flooding. Based on satellite imagery, and a sedimentological and geomorphological analysis, Komori et al. (2012) attributed an outburst from the supra glacial lake on the Tshojo glacier to the event. The most recent GLOF took place on 28 July 2015, when the Lemthang

Tsho outburst, discussed by Gurung et al. (2017), released an estimated 0.37 million $m^3$ of water downstream and heavy rainfall triggered the event.

Due to the significant risk it poses to downstream settlements in the event of a GLOF, the Thorthomi glacial lake has become a cause of serious concern. To address this issue, the Bhutan government initiated a high-priority project under the United Nations Framework Convention on Climate Change (UNFCCC) funding scheme in 2006, referred to as the National

Adaptation Plan of Action (NAPA). The project aimed to reduce the GLOF risk from Thorthomi Lake and involved lowering the lake's water level over four years, resulting in a reduction of 3.68 metres. However, due to challenging working conditions and health issues, the project fell 1.32 metres short of its target, although approximately 17 million $m^3$ of lake water was artificially released. The project additionally included setting up a GLOF Early Warning System along the Punakha-Wangdue Valley for alerting residents in the event of a GLOF. Several factors, including (1) rapid expansion of the Thorthomi supra

glacial lake, (2) the size of glaciers and probable future lake size, (3) the weakened left lateral moraine of the lake due to the 1994 Luggye GLOF, (4) active sliding on the moraine wall separating the Thorthomi and Rapstreng Lakes, (5) seepage from the lake, and (6) rock and snow avalanches, as summarised by Karma (2013), were taken into account in order to identify potential risks from Thorthomi Lake.

Although GLOF research and studies have gained global momentum in recent years, only a few studies have been performed

in Bhutan. Numerous studies have been conducted in Nepal and China that have simulated and assessed GLOF risks. Still, detailed studies on Bhutan's exposure to GLOF-related hazards are scarce and can be attributed to a lack of required field data, as well as Bhutan's limited exposure to the global scientific community. Warming climate exacerbates the hazards of GLOFs. Since such risk will only intensify in the coming years, there is an urgent need for a comprehensive GLOF assessment. Therefore, a study assessing hazards associated with glacial lakes and GLOFs is crucial for understanding hazards, as well as

their subsequent impacts on hydrological and socio-economic aspects within the Punatsangchhu Basin.

## 1.2 The focus of this study

We evaluated the potential risk of a GLOF from Thorthomi Lake, the largest of six potentially hazardous glacial lakes located within the Phochhu Basin. The physically based mathematical dam breach model, BREACH, was used to simulate the glacial lake dam breach and was coupled with HEC-RAS in order to route the flood wave propagating downstream. We sought to

simulate both the spatial extent and the lead time of flood wave arrival at several locations along the river. Our study is one of





the few studies that has simulated probable floods from Thorthomi Lake, and that has estimated inundation extent and flood arrival times within a scientific setting. Such studies form an essential basis for flood risk assessments, early warning system (EWS) installation, economic planning, countermeasure planning, designing, and stakeholder education and awareness programs.

Our study consisted of three parts:

    1)   estimating the geometry and water volume of Thorthomi Lake, a potentially hazardous glacial lake;

    2)   estimating the potential outburst flood hydrograph using a dam breach model, BREACH, and

    3)   assessing the GLOF hazard and potential risk using a 2D hydraulic model.

1.3 Structure of the paper

The paper contains seven sections. Section 1 introduces the overall concept of a Glacial Lake Outburst Flood (GLOF), and provides information obtained from previous studies and information related to GLOFs in the context of Bhutan. Section 2 describes the area of the study and the rationale behind our research within the focused area. Sections 3 and 4, respectively, provide a dam breach model and the hydrodynamic model used in this study. To verify our methods, Section 5 reproduces the 1994 Luggye GLOF event. Section 6 predicts a GLOF from Thorthomi Lake, and the consequences such a GLOF will cause.

A conclusion of our study is presented in Sect. 7.

   A schematic diagram of the data, models, methods, and process flow employed for our study is provided in Fig. 2.

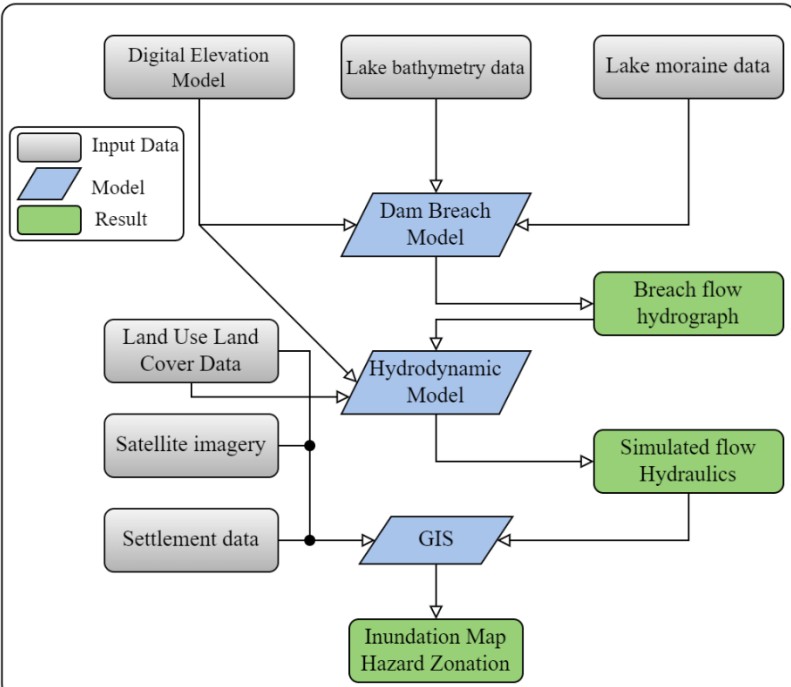

**Figure 2.** A schematic diagram of the methodology employed for our study. The input data used for the study are indicated with grey boxes, models used are indicated with blue boxes, and the results are shown in green boxes.





## 2 Study area: The Punatsangchhu River Basin

We assessed the flood risk focussed on the Phochhu catchment within the Punatsangchhu River Basin (PRB) caused by a GLOF from Thorthomi Lake. The basin located within the central portion of Bhutan (Fig. 3 (a)) is one of the largest basins in Bhutan, spanning an approximate area of 9,760 km$^2$; and is drained by two major rivers, the Phochhu and Mochhu Rivers (Fig. 3(a)) to the Indian plains. The PRB consists of five districts: the Gasa, the Punakha, the Wangdue Phodrang, the Dagana, and the Tsirang, spanning approximately 25% of the total area of the country (38,394 km$^2$). These districts constitute 16.6% (735,533) of the total population of Bhutan (NSB, 2018). The annual average discharge of the basin ranges from 194 to 374 m$^3$s$^{-1}$, with the highest recorded discharge of 2,654 m$^3$s$^{-1}$ occurring in 2009 during Cyclone Aila, as observed at the WangdiRapid station (location shown in Fig. 3 (c)).

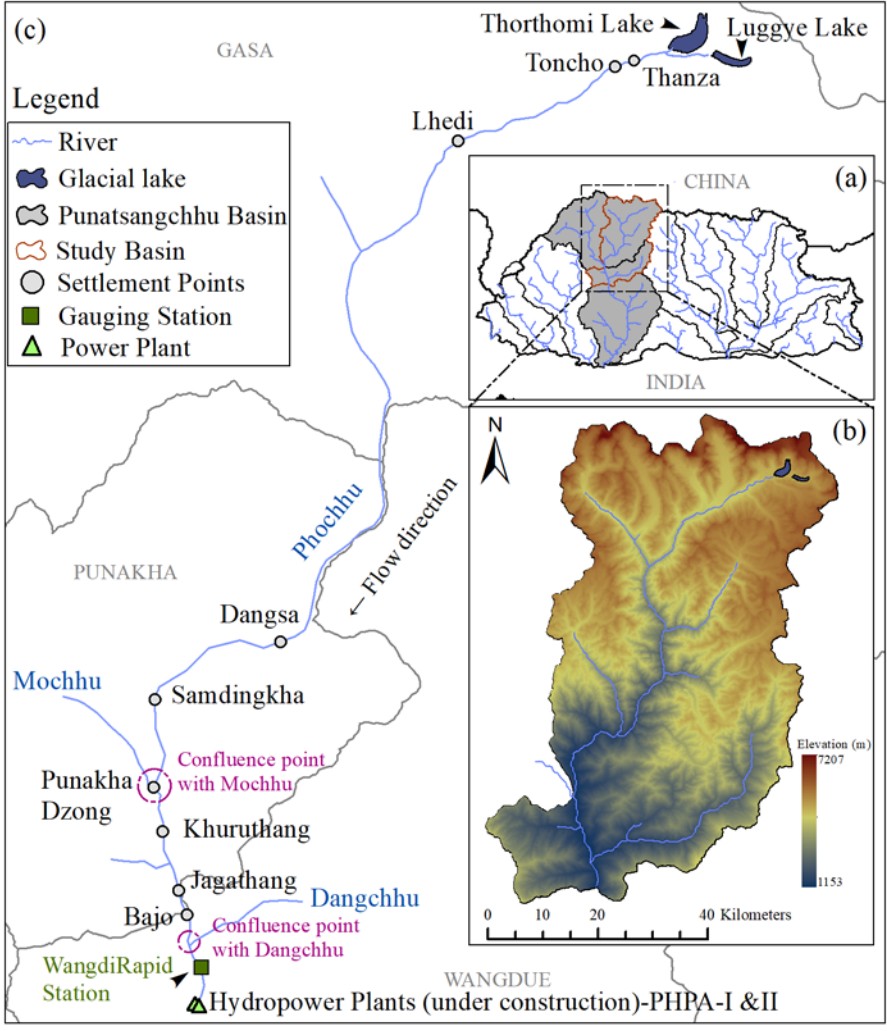

**Figure 3.** Map of the study area. (a) A Bhutan map showing rivers and river basins. (b) The elevation distribution of the study area. (c) The Phochhu River and major settlement points along the river.



The PRB is home to eleven (11) potentially dangerous glacial lakes. Nine (9) of these lakes lie within the Phochhu sub-basin. The Thorthomi glacial lake has an area of 4.3 km$^2$ (NCHM, 2019b), is located at the headwater of the Phochhu sub-basin at

over 4,440 meters above sea level and is considered to be one of the most dynamic and dangerous glacial lakes within Bhutan. The Phochhu River, one of the main tributaries of the Punatsangchhu River, originates from the high mountains of Lunana, in northern Bhutan, and flows some 90 kilometres downstream, where it joins Mochhu at Punakha Dzong (monastery) (Fig. 3 (c)), and flows from this area as the Punatsangchhu River. The Thorthomi glacial lake is widely recognized as a likely consequence of climate warming and is expanding each year, as feeding glaciers terminating in the lake rapidly melt. Based

on a comprehensive analysis of cryospheric, geotechnical, and geomorphological factors, Rinzin et al. (2023) concluded that Thorthomi Lake is highly susceptible to GLOF events.

The Punakha and Wangdue Phodrang districts within the Punatsangchhu River Basin (PRB) are renowned as Bhutan's primary rice production regions, contributing 16% and 11%, respectively, to the nation's total rice output (NSB, 2021). The area is also rich in historical and cultural heritage, with notable landmarks such as Punakha Dzong, which served as the former capital of

Bhutan. Located on the banks of the Phochhu and Punatsangchhu River are fertile flood plains, and the region encompasses settlements such as Samdingkha and Jagathang, together with major towns such as Khuruthang and Bajo. The floodplain of the Punatsangchhu River accommodates these settlements, while downstream, approximately 115 kilometres away from Thorthomi Lake, two significant hydropower plants, the Punatsangchhu Hydroelectric Project Authority - PHPA-I and II, are currently under construction. Given the exposure of critical infrastructure and settlements to potential GLOFs from the lake,

an assessment of hazards within this area is of paramount importance.

## 3 The dam breach model of the moraine dam

### 3.1 Previous studies regarding the moraine dam breach

GLOFs are triggered by a breach in the moraine dam that holds the lake in place and are caused by an external triggering event. While the structure of the dam itself is an important factor, the destabilisation of the dam due to a trigger event is the primary

cause of a breach.

To estimate the potential flood flow resulting from a dam breach and associated hazards, several studies have simulated dam breach floods using dam breach models (Bajracharya et al., 2007; Hagg et al., 2021; Huggel et al., 2002; Koike & Takenaka, 2012; Maskey et al., 2020; Meyer et al., 2006; Shahrim & Ros, 2020; Wang et al., 2008; Worni et al., 2014). BREACH is a numerical model describing the dam breach process and the resulting outflow hydrograph. The model is based on fundamental

principles of hydraulics, sediment transport, soil mechanics, and the physical properties of dam materials and the reservoir. The model is physically based and is designed to predict the size, shape, and time of formation of a breach in a dam, as well as the resulting flow rate and the volume of water released. The BREACH model has been widely used in studies of dam breach flood hazards and risk assessments (Fread, 1988).





Koike and Takenaka (2012) used the BREACH model coupled with the flood flow model, FLO-2D, in order to perform a
scenario analysis on the risks of GLOF on the Mangdechhu River Basin due to an outburst of the Metatshota glacial lake in
Bhutan. The study concluded that although the breaching potential of the lake is low due to the wide crest and gentle slope of
the moraine dam, a GLOF would affect several houses and farmland located on the flood plain (Koike & Takenaka, 2012).
Hagg et al. (2021) performed a GLOF hazard assessment within the Mochhu Basin in Bhutan using the HEC-RAS dam break
module, simulating a dam breach of the Shintaphu glacial lake, and concluded that the risk is comparably small.

Our study used BREACH for describing the dam breach process for target lakes. Unlike parametric models, physically based
breach models consider the geotechnical aspects of dam materials, as well as hydraulic and sediment transport (Fread, 1988;
Maskey et al., 2020; Worni et al., 2014), which increases the predictive accuracy of future GLOF processes. The dam is
assumed to breach due to overtopping flow resulting from a trigger event such as an ice calving/avalanche or a rock avalanche.
Most of the geotechnical properties of the dam materials required as an input parameter are available in the report published
by the National Center for Hydrology and Meteorology (NCHM, 2019a). A few were published by Koike and Takenaka,
(2012). Some unavailable data was estimated by referring to previous studies.

## 4 Flood routing

### 4.1 The hydrodynamic model

A hydrodynamic model is essential for understanding the characteristics of a flood wave caused by a GLOF propagating
downstream, as well as the associated flood characteristics required to ascertain potential risks from a flood.
Numerous studies, such as the Hydrologic Engineering Centre's-River Analysis System (HEC-RAS) used in Maskey et al.
(2020), have employed various hydrodynamic models in order to simulate the propagation of outflow from glacial lake
breaches in Nepal. Similar approaches that couple dam breach models with hydrodynamic models ( e.g., Bajracharya et al.,
2007; Koike & Takenaka, 2012; Westoby et al., 2015; Worni et al., 2014) have been performed in order to model the GLOF
process chain in various regions.
Worni et al. (2014) provided a summary of various hydrodynamic models that have been used to model GLOFs. Discussed
models include HEC-RAS, FLO-2D, BASEMENT, and Delft3D. The choice of a hydrodynamic model depends on factors
such as the end objective, data availability, and the available budget. Each model has its own level of accuracy; however, the
accuracy of results is primarily dependent on the precision of the elevation model, including channel geometry and floodplain
topography. Errors in the elevation model can lead to inaccuracies in results (Casas et al., 2006; Xu et al., 2021).
The Hydrologic Engineering Centre's River Analysis System (HEC-RAS) is a commonly used hydrodynamic model that
allows users to perform 1D & 2D steady/unsteady flow simulations (Brunner & CEIWR-HEC, 2016). We used the HEC-RAS
to perform a 2D unsteady flow simulation of floods caused by a glacial lake dam breach. Since they represent spatially varied
flood hydraulics (Horritt & Bates, 2001), the two-dimensional models employed are standard in flood modelling. In a 2D
unsteady simulation, flow varies in time, along two spatial dimensions, and such processes are predicted by the laws of





conservation of mass (continuity) and the conservation of momentum for two horizontal directions. We used a full momentum equation set (the shallow water equations) in order to simulate flooding as clear water flow. Although high viscosity and hyper-concentrated (sediment entrained) flows are inherent to the GLOF phenomenon (Clague & Evans, 2000; Vuichard & Zimmermann, 1987), to simplify modelling complexity and data requirements, most studies (Hagg et al., 2021; Koike &

Takenaka, 2012; Maskey et al., 2020; Rinzin et al., 2023) have simulated GLOFs as clear water flow.

Other important considerations in hydrodynamic modelling are Manning's roughness coefficient and channel geometry. Both have significant impacts in predicting inundation extent and flow characteristics (Mosquera-Machado & Ahmad, 2007; Ye et al., 2018; Zhu et al., 2019). Hagg et al. (2021) demonstrated the influence of Manning's roughness coefficient for glacial lake outburst floods from Shintaphu glacial lake and the Mochhu Basin, Bhutan; and concluded that channel roughness is not

essential for inundation extent but exerts a significant effect on flood velocity and flood arrival time.

## 4.2 The ground elevation distribution

### 4.2.1 Available data sources

The accuracy of hydrodynamic model results is heavily influenced by the quality of the elevation model used and is crucial for a precise representation of the terrain for flood inundation modelling (Gyasi-Agyei et al., 1995; Yamazaki et al., 2014,

2017). Casas et al. (2006) demonstrated the effects of a topographic data source and resolution on flood peak discharge and the extent of inundation, and then concluded that laser-based elevation data is a suitable source for hydraulic modelling. Similarly, the influence of grid size on inundation propagation and water depth under varied topographical settings in 2D modelling has been demonstrated (Tsubaki & Kawahara, 2013); and has indicated that both the fine grid size representing main topographic features of the floodplain and accurate elevation at each grid point are essential for simulating flood flow

with less uncertainty. Therefore, using the best available elevation model for the hydrodynamic simulation of floods is essential. Since an accurate elevation model is essential for accurate hydrodynamic simulations, elevation models include error in varying degrees. To manage elevation error, various methods for correcting generic noise errors and bias have been proposed and have been used in elevation models, prior to running hydrodynamic/hydrological analyses. The Multi-Error-Removed Improved-Terrain (MERIT) Hydro DEM (see Fig. 4 (c)) was developed based on SRTM and AW3D DEM, and water layer

data at a 3-arc sec resolution (~90 m) was developed for a river hydrology analysis at global, as well as at local, scales (Yamazaki et al., 2017, 2019).

Other bias-corrected elevation data, including the Forest And Building removed Copernicus DEM (FABDEM) (see Fig. 4 (b)), developed from Copernicus DEM (COPDEM), where the height of trees and buildings are removed using machine learning, is also a preferable source for terrain data (Hawker et al., 2022).

In this study, the AW3D Digital Surface Model (DSM) obtained from the Remote Sensing Technology Centre (RESTEC) of Japan was utilized as the primary source of topography. This DSM was jointly developed by the RESTEC and NTT DATA Corporation, utilising PRISM data acquired by the Advanced Land Observing Satellite (ALOS) of the Japan Aerospace


Exploration Agency (JAXA). The cell size of the DSM used was 2.2 metres for both the *X* and *Y* directions and was projected to WGS84-UTM-Zone 45N. The DSM was selected because it provided a finer terrain representation than other freely

available Digital Elevation Models (DEMs). Figure 4 illustrates the difference in terrain representation using three terrain data sources. The AW3D-2.2 m DSM represents topography much better than the other two.

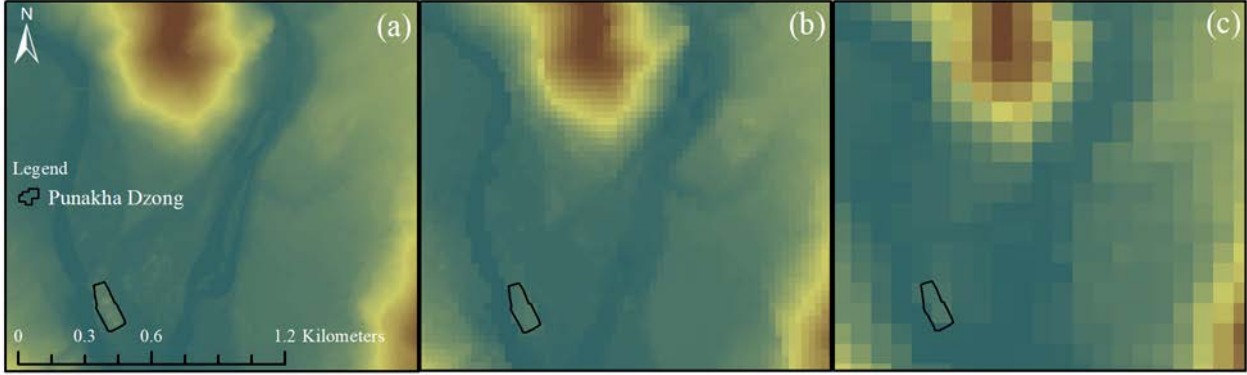

**Figure 4.** The Punakha Dzong region as represented by three different terrain models: (a) the AW3D-2.2 m DSM, (b) the FABDEM-30 m, and (c) the MERIT HydroDEM-90 m.

**4.2.2 Elevation errors and their correction**

Since the AW3D DSM was obtained using satellite photogrammetry, representation of the river bottom, especially in forested and deep gorge areas, is sometimes inaccurate. To avoid anomalies in the elevation model and the hydrodynamic simulation results, a river channel delineation was performed. The presence of spikes within the DSM, along the river's path, can obstruct the downstream flow of floodwater, resulting in the formation of non-existent deep pools. One way to improve topography

surrounding a river is the use of bathymetric survey data. However, no such survey has been conducted within the study area. To improve representation for the river channel, our study utilised a rule-based correction method.

The Agriculture Conservation and Planning Framework (ACPF) is a GIS-based tool developed by the United States Department of Agriculture (USDA) to identify areas with impeded water flow, and to improve hydrologic flow using flow direction and an accumulation analysis (Porter et al., 2016). While the ACPF is a valuable tool for hydrologic flow and

watershed planning, it has limited applicability for terrain correction in hydrodynamic modelling because the ACPF does not allow users to define the bathymetry of a river channel.

Another widely used channel modification method is the in-built function of HEC-RAS. The Channel Design/Modification Editor tool is a module used to modify an unrealistic cross-section or to introduce a user-defined channel cross-section (Brunner, 2016). The tool effectively removes spikes in the elevation model along a river channel while maintaining the natural slope of

the represented topography. The modified channel TIN (Triangulated Irregular Network) can be overlain on the original DSM and exported as a single raster file with modified features. Rinzin et al. (2023) applied the method in order to modify terrain and to delineate river flow paths for a GLOF simulation. For our study, we used this tool to condition the DSM in the middle





region of the model domain, a forested deep gorge, where huge spikes were included within the elevation model. The modification was only applied to the channel section. The remaining portion was left as it was.

## 5 Reconstruction of the 1994 Luggye GLOF

### 5.1 Description of the event

Luggye glacial lake is one of the potentially dangerous glacial lakes in Lunana, Bhutan's northern region. As shown in Fig. 5, the lake is one of four glacial lakes in an area that span a few kilometres and had an outburst in 1994. Although there is no detailed official documentation on the GLOF at Luggye glacial lake, reports and articles describing the event do exist (eg., Koike & Takenaka, 2012; Meyer et al., 2006; Richardson & Reynolds, 2000; Watanabe & Rothacher, 1996). The event was also documented in a technical report (Leber et al., 2000), when the Royal Government of Bhutan launched a major investigative project in 2000 in order to study the cause of the event.

The 1994 GLOF was a cascading phenomenon, where sudden drainage of the upstream Druk Chung glacial lake (see Fig. 5) into Luggye Lake increased hydrostatic pressure on the moraine dam of Luggye Lake, releasing 18 million $m^3$ of flood water (Leber et al., 2000). The GLOF claimed the lives of 21 people, inflicted major damage to infrastructure and settlements downstream; notably, the Punakha Dzong (monastery) suffered significant damage, although it is located 93 kilometres downstream from the lake (Richardson & Reynolds, 2000; Watanabe & Rothacher, 1996). A peak flow rate of 2,539 $m^3\,s^{-1}$ was observed at the WangdiRapid gauging station, located 15 kilometres downstream of Punakha Dzong (see Fig. 3) and approximately 108 kilometres downstream of the flood source (data from NCHM.) Here, small contributions from the Mochhu and Dangchhu basins, as well as other small tributaries, affected the peak flow rate.

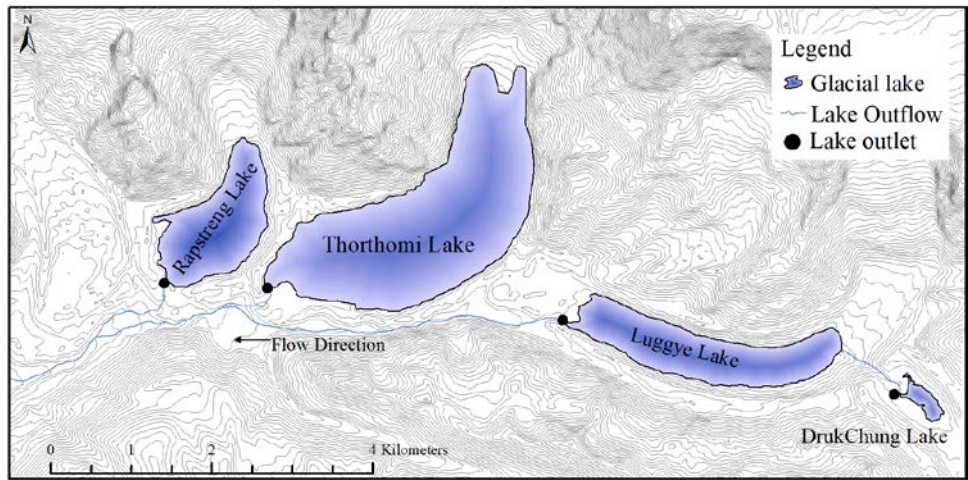

**Figure 5.** Four glacial lakes within the Lunana region (Lunana Complex).





To estimate the breach outflow hydrograph, several studies have attempted to reconstruct the 1994 Luggye GLOF event (e.g., JICA, 2001; Koike & Takenaka, 2012; Meyer et al., 2006). JICA (2001), and Koike and Takenaka (2012), estimated that peak discharge from the Luggye Lake breach ranged from 1,800 to 2,500 m$^3$s$^{-1}$, depending on inflow conditions measured by Yamada et al., (2004). In this study, to verify our methods prior to analysing the Thorthomi GLOF event, we reconstructed the 1994 GLOF event. Main aspects used for verification included: (1) the BREACH model, (2) DSM error correction methods,

and (3) Manning's coefficient.

### 5.2 Methods

### 5.2.1 The breach model

For reconstruction of the 1994 Luggye GLOF, the dam breach outflow hydrograph was estimated using BREACH (Fread, 1988). The bathymetry of Luggye Lake (Fig. 6) and the material properties of the moraine dam (see Table 1) required for the

model were based on various reports (NCHM, 2019a, 2019b). Topographic data of the moraine dam was derived from the Digital Surface Model (DSM). Since wave overtopping is a more common failure mode for moraine-dammed glacial lakes as compared to a piping failure (Neupane et al., 2019), to estimate breach outflow from Luggye Lake, overtopping failure of the moraine dam was assumed. The properties of moraine dam material have a significant effect on the growth of a breach (Maskey et al., 2020; Westoby et al., 2014). The mechanism in which the formation of a breach largely occurs determines the shape of

the breach outflow hydrograph (Westoby et al., 2014). Therefore, gathering accurate in-situ data for reliable breach process reproduction is essential.

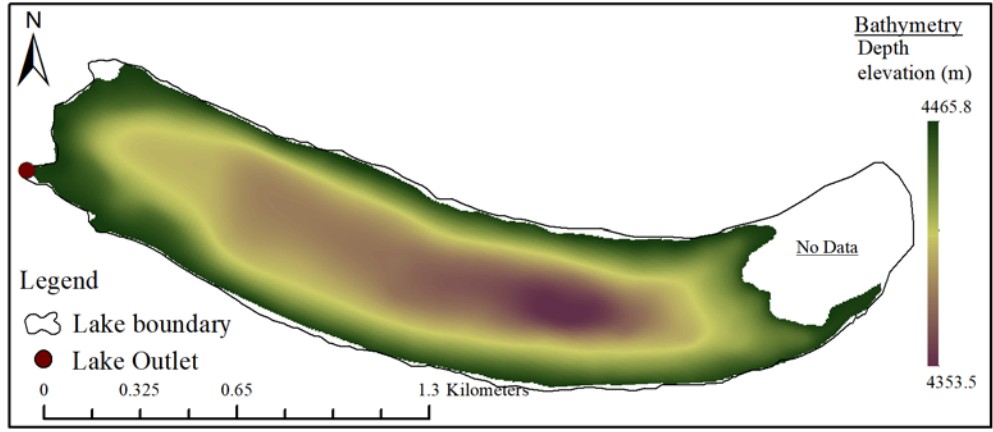

**Figure 6.** The bathymetry of Luggye Lake (data from NCHM).

Based on the estimation by Fujita et al. (2008), deduced from a combination of field measurements and remote sensing observations, the level of lake water was reduced 19 metres during the event. Table 1 provides moraine dam data and the





moraine material properties of Luggye Lake used for the dam breach model. Parameters were either estimated or were obtained from various available reports and research papers.

**Table 1.** Input parameters for the Luggye Lake BREACH model

| Moraine Dam Data | | Dam Material Properties | |
|---|---|---|---|
| Surface area of the lake (km$^2$) [RSA] | 1.46 [b] | Grain size ($D_{50}$) (mm) | 1.362 [b] |
| Volume of water in the lake (mil. m$^3$), Eq. (4) | 65.19 [b] | Porosity (%) | 36.5 [c] |
| Maximum depth of the lake (m), Eq. (3) | 96.93 [b] | Cohesive strength (kN m$^{-2}$) | 1.5 [b] |
| Top elevation of the dam (m) [HU] | 4465 [a] | Internal friction (degree) | 41 [b] |
| Toe elevation of the dam (m) [HL] | 4370 [a] | Unit weight (kN m$^{-3}$) | 22.92 [b] |
| Slope of the upstream face of the dam (1: ZU) | 1:4.8 [a] | Manning's coefficient (s m$^{-1/3}$) | 0.07[b] |
| Slope of the downstream face of the dam (1: ZU) | 1:6.5 [a] | | |

[a] Estimated in this study, [b] NCHM (2019a), [c] Koike and Takenaka (2012)

### 5.2.2 The hydrodynamic model

Downstream propagation of the flood was simulated using the HEC-RAS model. The calculation domain was defined by the 2D flow area. The overall size of the flow area was 64 km$^2$. The domain was modelled at a 20-m-resolution computational

grid, consisting of 157,188 computational cells, and solved with a time step of one second. The elevation of each grid cell was specified based on a 2.2 m, hydro-conditioned digital surface model (DSM, see Sect. 4.2.1), and Manning's $n$ was set to 0.35, in the range provided by the HEC-RAS manual (Brunner, 2016). The dam breach outflow hydrograph obtained from the BREACH model was used as the upstream boundary; and normal depth, calculated based on downstream slopes derived from the DSM, was used as the downstream boundary condition.


### 5.3 Results

### 5.3.1 The dam breach hydrograph

As shown in Fig. 7 (a), simulated peak flow ($Qp$) of the dam breach outflow hydrograph was 6,030 m$^3$ s$^{-1}$, and 1.9 hours (~114 minutes) were required to reach peak flow (the time to peak, $Tp$). The volume of the GLOF and the reduction of lake water

level due to the event were 21 million m$^3$ and 19 m, respectively, in agreement with the findings of Fujita et al. (2008). Dimensions for the estimated breach of the dam are provided in Fig. 7 (b).



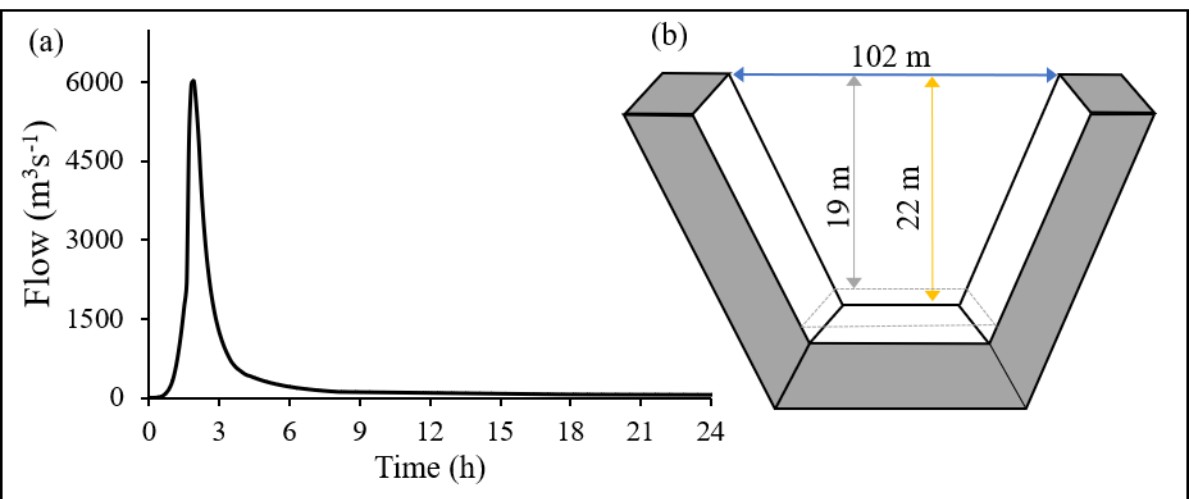

**Figure 7.** (a) A breach outflow hydrograph from the BREACH model; and (b) an illustration of breach parameters, breach width (↔), breach depth (↕), and the change in water surface elevation (↕).

### 5.3.2 Hydrodynamic processes

Flow hydrographs at various locations along the flow path are provided in Fig. 8. Conforming to the findings of Meyer et al. (2006), after approximately 6 hours, the GLOF had a peak discharge of 2,897 $m^3 s^{-1}$ as it reached Punakha Dzong, located 93 kilometres downstream of the lake. Peak flow at the WangdiRapid Station (shown in Fig. 3(c)), 15 kilometres downstream of Punakha Dzong, was 2,455 $m^3 s^{-1}$, close to the recorded value of 2,539 $m^3 s^{-1}$. Here, the recorded flow rate included the contribution of normal flow from tributaries, which was not accounted for in our analysis. Good agreement of results for simulated flow and flood travel time with observed data, as well as previous studies, indicated that the performance of the employed models, and the modelling approach, is adequate and is capable of yielding satisfactory results for predictive modelling of the target lake. Total inundated area along the basin was approximately 13.1 $km^2$.

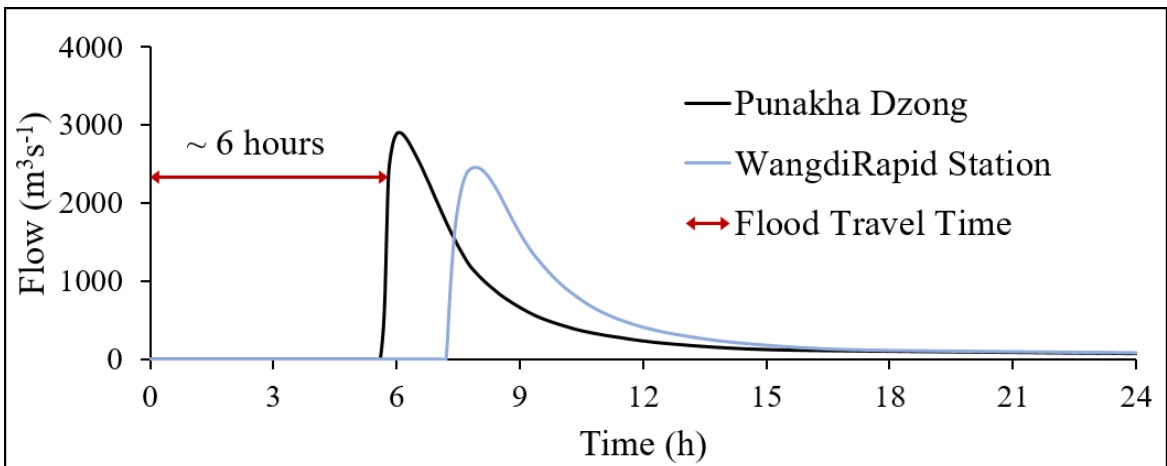

**Figure 8.** A simulated GLOF hydrograph at different locations along the flow path.





## 6 Prediction of the Thorthomi GLOF

### 6.1 An outline of methods for prediction

We considered the failure of the moraine dam for the existing outlet (see Fig. 9) due to an overtopping flow induced by any probable triggering event. To estimate the potential volume of the lake, the bathymetry of the Thorthomi glacial lake is needed. Although several literature estimates of lake volume are available (Karma, 2013; Singh, 2009), no details on how volumes were estimated have been documented. Maximum lake depth and volume for Lake Thorthomi were estimated based on the parametric relationship proposed by Sakai (2012). Bathymetry was approximated using an idealised geometric shape, by

employing the area-volume relationship conceptualised by Cook and Quincey (2015). A dam breach was simulated using the BREACH model. The breach outflow hydrograph was routed downstream using the HEC-RAS model.

### 6.2 Bathymetry of the Thorthomi glacial lake

#### 6.2.1 Limited bathymetry survey data

Glacial lakes are generally formed from a depression left behind by retreating glaciers, which, in most cases, are produced
when a moraine is filled with melt water. Depending on the geomorphological condition of the land, the presence of sediment, and glacial over-deepening capacity, formed glacial lakes can manifest specific lake bathymetry and influences on glacial hydrology (Cook & Swift, 2012). Due to their remote locations and high elevations, accessing and conducting field surveys in order to map glacial Lake Bathymetry is challenging.

Measurements of lake bathymetry are crucial in determining a lake's volume and surface area, which are necessary for
assessing potential flood volumes (PFV) and the risk of GLOFs. In 2019, the National Centre for Hydrology and Meteorology (NCHM) conducted bathymetric surveys in 14 of the 25 identified potentially dangerous glacial lakes and mapped their maximum depth and volume. However, due to the difficulty associated with conducting a survey, the bathymetry of the Thorthomi glacial lake remains unknown, even though the lake is considered to be a critical lake that could burst in the future. Since lake geometry is a crucial parameter for dam breach modelling and subsequent hydraulic routing, lake depth and volume
are estimated using the procedures described below.

#### 6.2.2 Regression analysis for the lake geometry estimation

Estimating the potential flood volume of a glacial lake is critical in determining the magnitude of a GLOF. However, due to the challenging and inaccessible environments in which glacial lakes are often located, bathymetry data, which is necessary for calculating the volume of a lake, can be scarce. To address this issue, various studies have proposed methods for estimating
glacial lake depth and volume based on other more accessible parameters such as lake area (S. J. Cook & Quincey, 2015; Huggel et al., 2002; O'Connor et al., 2001; Sakai, 2012), as well as the depression angle from the lakeshore (K. Fujita et al., 2013) and the surrounding topography (Heathcote et al., 2015). Empirical relationships such as area-volume and area-depth are useful for estimating a lake's depth and potential flood volume. Cook and Quincey (2015) refined the area-volume


relationship proposed by Huggel et al. (2002) by increasing the sample size and removing duplicate sample data. They also

classified the predictability of lake volume and depth based on regions and lake types and determined that predictability is

influenced by a lake's origin and evolution. The relationship proposed by Cook and Quincey (2015) is based on 45 data points,

including the data points used by Huggel et al. (2002) for removing duplicate data, and takes the following form:

$$D_{\text{mean}} = 0.1697\,A^{0.3778}, \tag{1}$$


where $D_{\text{mean}}$ is the mean depth (in metres) and $A$ is the area (in square metres). The volume-area relationship ($V$, volume in

cubic metres) can be derived by multiplying the area by both sides, as follows:

$$V = 0.1697\,A^{1.3778}. \tag{2}$$


A similar approach was proposed by Sakai (2012), where maximum depth was taken into consideration rather than mean depth.

The bathymetric measurement data of 17 glacial lakes (15 moraine-dammed glacial lakes and 2 thermokarst lakes) from Bhutan,

Nepal, and Tibet were used to derive an area-maximum, depth-volume relationship in order to enable estimations of depth and

volume from the area of glacial lakes (Sakai, 2012). The regression equation takes the following form:


$$D_{\text{max}} = 95.665\,A^{0.489}, \tag{3}$$

where $D_{\text{max}}$ is maximum depth (in metres), $A$ is the area (in square kilometres), and the volume-area relationship (V, volume

in million m$^3$) takes the following form:


$$V = 43.24\,A^{1.5307}. \tag{4}$$

Unlike previous studies (e.g., Cook & Quincey, 2015; Fujita et al., 2013; Huggel et al., 2002; O'Connor et al., 2001), where

mean depth is estimated, which is straightforward for estimating volume, Eq. (3) estimates maximum depth. We used the

equations proposed by Sakai (2012) in order to estimate the maximum depth and volume of the Thorthomi glacial lake. The

equations allow the independent calculation of maximum depth and volume. As conceptualised by Cook and Quincey (2015),

bathymetry of the lake was estimated based on the idea of idealised geometric shape. The lake bottom was also assumed to

follow an elliptical shape, as commonly observed in most moraine-dammed glacial lakes in Bhutan.





### 6.3 The breach model

Breach initiation is assumed to occur due to an overtopping wave. Moraine material properties and the topographic data of a moraine dam are either estimated from available terrain data or adopted from available reports and research documents. Table 2 provides moraine dam data and moraine material properties for Thorthomi Lake used in the BREACH model.

**Table 2.** Input parameters for the Thorthomi Lake BREACH model

| Moraine Dam Data | | Dam Material Properties | |
|---|---|---|---|
| Surface area of the lake (km$^2$) [RSA] | 4.3 [b] | Grain size ($D_{50}$) (mm) | 2.01 [b] |
| Volume of water in the lake (mil. m$^3$), Eq. (4) | 400 [a] | Porosity (%) | 36.5 [c] |
| Maximum depth of the lake (m), Eq. (3) | 161 [a] | Cohesive strength (kN m$^{-2}$) | 1.5 [b] |
| Top elevation of the dam (m) [HU] | 4446 [a] | Internal friction (degree) | 39 [b] |
| Toe elevation of the dam (m) [HL] | 4370 [a] | Unit weight (kN m$^{-3}$) | 22.43 [b] |
| Slope of the upstream face of the dam (1: ZU) | 1:6.2 [a] | Manning's coefficient (s m$^{-1/3}$) | 0.07[b] |
| Slope of the downstream face of the dam (1: ZU) | 1:6.3 [a] | | |

[a] Estimated in this study, [b] NCHM (2019a), [c] Koike and Takenaka (2012).

### 6.4 The hydrodynamic model

We used the same hydrodynamic model from Section 5. Downstream propagation of the flood was simulated using the HEC-RAS model. The calculation domain is defined by the 2D flow area. The overall size of the flow area was 62 km$^2$. The domain was modelled using a 20 m resolution computational grid consisting of 153,790 computational cells with a temporal resolution
of 1 second.

### 6.5 Results of the Thorthomi GLOF prediction

#### 6.5.1 Lake bathymetry

**Table 3** provides the estimated maximum depth and volume of the PDGLs using Eqs. (3) & (4), as proposed by Sakai (2012).

**Table 4**. List of potentially dangerous glacial lakes with measured lake volume and maximum depth, and estimated values using the relationship derived by Sakai (2012).

| | Measured value (NCHM, 2019a) | | | Sakai (2012) | |
|---|---|---|---|---|---|
| Potentially dangerous glacial lakes (PDGL) | Area (km$^2$) | Volume (mil. M$^3$) | Max. Depth (m) | Volume (mil. M$^3$) | Max. Depth (m) |

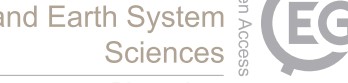

| | | | | | |
|---|---|---|---|---|---|
| Mo_gl 202 (Latshokarp) | 0.068 | 0.1 | 10 | 0.71 | 25.78 |
| Mo_gl 234 (Sintaphu tsho) | 0.238 | 6.4 | 54 | 4.81 | 47.44 |
| Pho_gl 84 | 0.742 | 9.28 | 37.39 | 27.38 | 82.69 |
| Pho_gl 148 | 0.637 | 26.3 | 101 | 21.68 | 76.75 |
| Pho_gl 163 (Tarina I) | 0.250 | 5.4 | 43 | 5.20 | 48.64 |
| Pho_gl 164 (Tarina II) | 0.446 | 13 | 67.5 | 12.56 | 64.48 |
| Pho_gl 209 (Raphstreng) | 1.242 | 54.6 | 110 | 60.20 | 106.36 |
| Pho_gl 210 (Lugge Tsho) | 1.46 | 65.19 | 96.93 | 77.11 | 115.11 |
| Mang_gl 99 (GLT 9) | 0.229 | 4.74 | 51.7 | 4.52 | 46.51 |
| Mang_gl 106 (Metatshota) | 1.2 | 41 | 120 | 57.12 | 104.58 |
| Mang_gl 270 (Zanam F) | 0.223 | 4.71 | 60.4 | 4.35 | 45.94 |
| Mang_gl 307 (Zanam B) | 0.862 | 37 | 103 | 34.43 | 88.971 |
| Mang_gl 310 (Zanam G) | 0.206 | 1.87 | 21.19 | 3.85 | 44.20 |
| Cham_gl 198 (Phudung lake) | 0.582 | 10.76 | 63 | 18.90 | 73.46 |
| Cham_gl 383 (Chubda Tsho) | 1.388 | 21.69 | 55.79 | 71.37 | 112.30 |
| Thorthomi Lake | 4.3 | - | - | 400 | 161 |

The estimated volume and maximum depth of Thorthomi Lake from Eq. (3) and (4) were 400 million m³ and 161 metres, respectively. The estimated volume and maximum depth of the Thorthomi Lake falls within the predicted band considering 95% confidence level. The equations used showed good relationship between area and volume and area and maximum depth, with the prediction range of 281 million m³ – 400 million m³ – 560 million m³ (lower bound – calculated value – upper bound) for volume prediction and prediction range for maximum depth lies between 130 meters – 161 meters – 270 meters. Compared

to other glacial lakes in Bhutan, the estimated parameters from Table 3 indicate that the Thorthomi glacial lake is one of the largest and deepest lakes. The bathymetry of Thorthomi Lake, estimated based on the above parameters, is provided in Fig. 9.

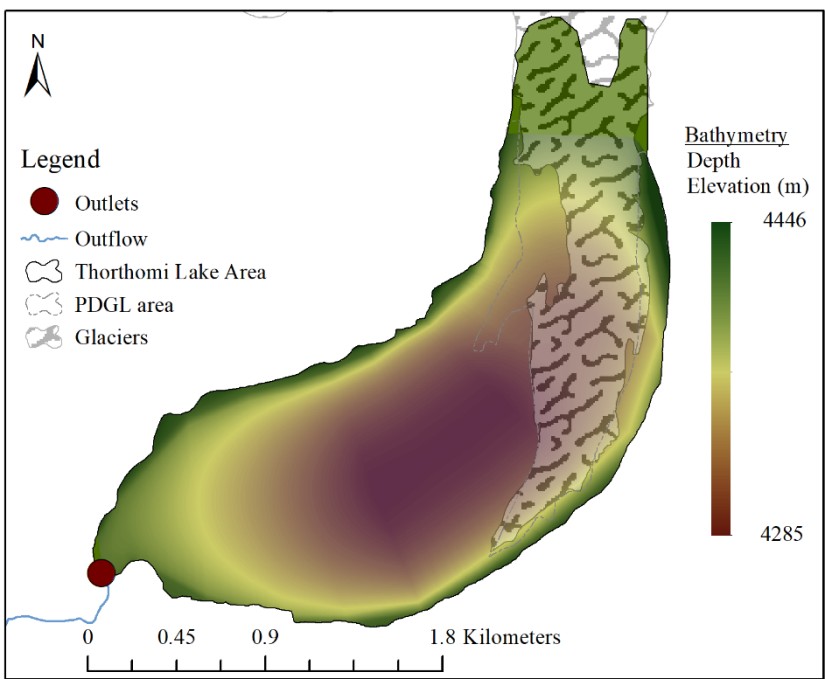

**Figure 9.** The estimated bathymetry of Thorthomi Lake.

### 6.5.2 The dam breach process

The breach was modelled using overtopping flow conditions. Different dam breach scenarios for maximum breach, and a partial breach for a half breach width and depth (50% of maximum width and depth) were simulated in order to ascertain potential risk under various breaching possibilities, including a partial dam breach which occurred in 1994 (e.g., the 1994 Luggye GLOF). Simulated peak flow ($Qp$) resulting from the Thorthomi dam breach under different breach scenarios ranged from 9,700 m$^3$ s$^{-1}$ (for a 50% breach depth) to 16,360 m$^3$ s$^{-1}$ (for maximum breach width and depth), with a time to peak ($Tp$) of 3.4 to 4 hours, as shown in Fig. 10 (a). The bathymetry of the lake and the topography of the moraine dam dictates the total lake draw down depth and the volume of the outburst flood. In this study, we estimated that 100 metres of lake water depth will be lowered before the breach outflow channel becomes sufficiently stable, after sending 283 million m$^3$ of flood water downstream (Fig. 10 (b)). The breach outflow channel was assumed to be stable when its bottom elevation reached its natural bed level of the downstream channel and down-cutting ceased.




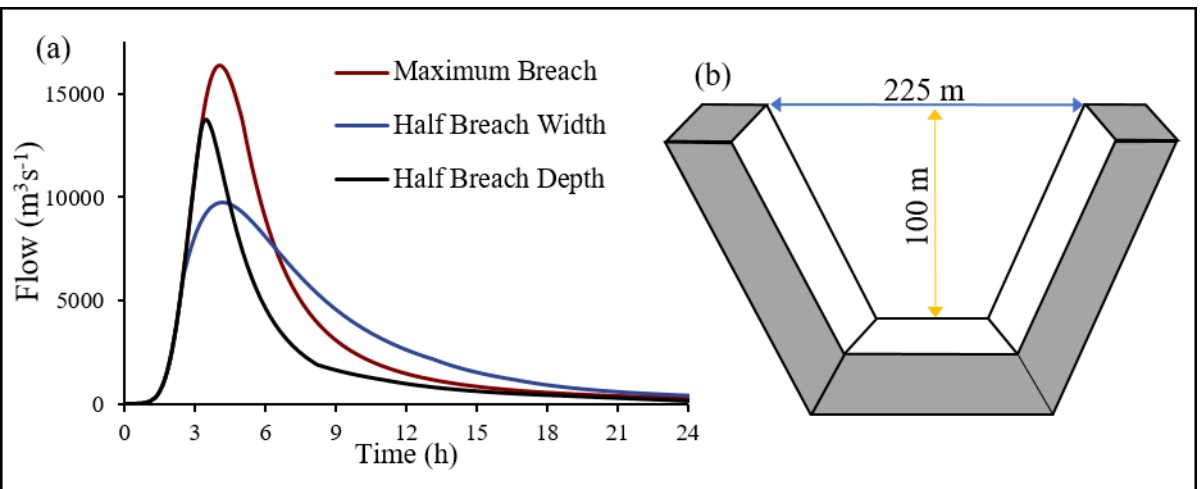

**Figure 10.** (a) A dam breach outflow hydrograph obtained from the BREACH model for three different scenarios; and (b) breach parameters, breach width (↔), and breach depth (↕) in metres for the maximum breach scenario.

### 6.5.3 Peak flow and flood travel time

The simulated flow hydrographs for three different scenarios at eight major settlement areas are provided in Fig. 11. Peak flow of the GLOF gradually attenuated as it propagated downstream. Peak flow at Punakha Dzong ranged from 8,900 to 14,100 $m^3 s^{-1}$, and decreased from 8,200 to 11,500 $m^3 s^{-1}$ when it arrived at the hydropower plant (PHPA-I).

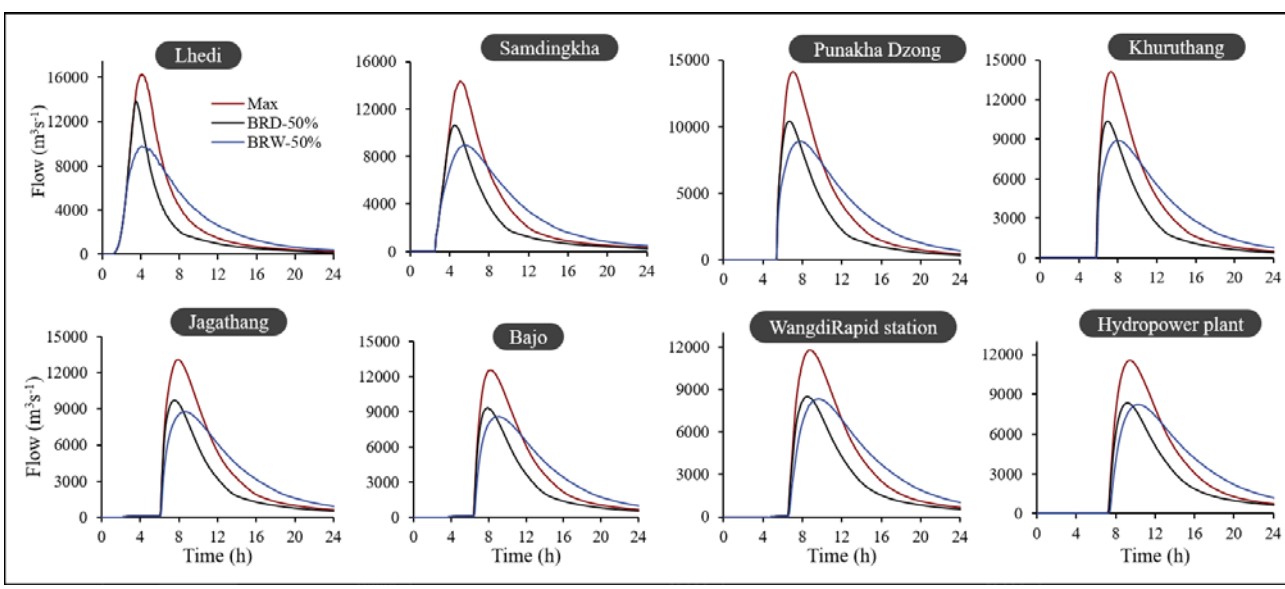

**Figure 11.** A simulated flow hydrograph at important locations, derived from the HEC-RAS result for each scenario. (Max: maximum breach, BRD-50%: half of maximum breach depth, BRW: half of maximum breach width).





Since such information is needed in order to estimate the area needed for evacuation and the lead time for evacuation, flood
travel time and peak flow are essential parameters for early warning purposes. Flood travel time in this study was calculated
based on the timing of the breach outflow hydrograph and the flow hydrograph at the point of interest, when there was a
significant inundation depth and extent. Peak flow is the maximum simulated flow resulting from the dam breach, derived
from the HEC-RAS simulation.

A schematic representation of an approximate distance, peak flow, averaged channel slope, and the estimated flood travel time
for a maximum breach condition is provided in Fig. 12.

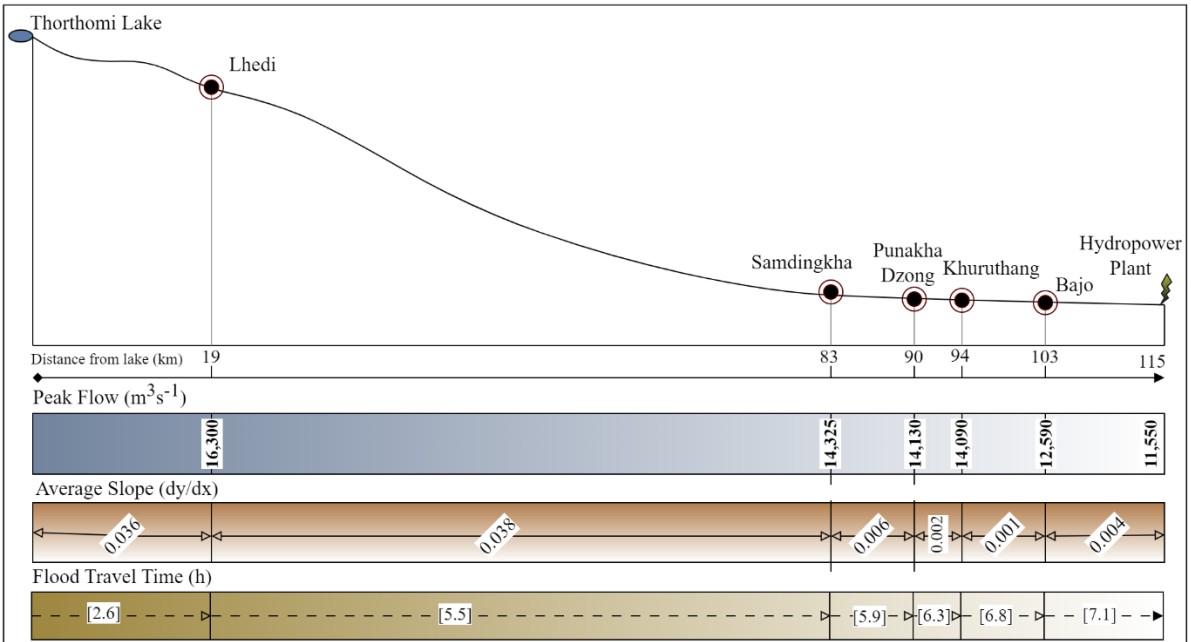

**Figure 12.** A schematic representation of flood parameters at six important locations along the flow path for the maximum breach scenario.

**6.5.4 Inundation mapping and hazard potential**

To perform further analysis and to produce inundation maps, flood depth simulated with HEC-RAS was exported to a GIS
(Geographic Information System). The flood depth distribution, highlighting the five vulnerable areas for a maximum breach
scenario, is provided in Fig. 13. The villages of Thanza, Toncho, and Lhedi, located in the northern-most part of the study area
(see Fig. 3), are expected to be inundated under a Thorthomi GLOF scenario. Major settlements along the river basin lie in the
lower valleys of the Punakha and Wangdue districts, where large areas are expected to be flooded. Major towns and settlements,
such as Samdingkha, Khuruthang, and Bajo, are expected to be inundated. The total inundated area due to a Thorthomi GLOF
with the maximum breach is approximately 22 km².

**Figure 13.** A maximum GLOF inundation map of the study area under the maximum breach scenario. Map data: Google Earth © 2023 CNES/Airbus, Maxar Technologies.



Figure 14 compares the maximum inundation depth and extent for three different scenarios for the town of Khuruthang. Simulation results for the three scenarios considered in this study revealed that the overall inundation extent and flood depths were higher for the maximum breach scenario. However, the depth and flood extent for the two other scenarios were comparable to the maximum breach scenario. The results indicate that even for a partial breach of the moraine dam substantial

damage within the downstream is expected.

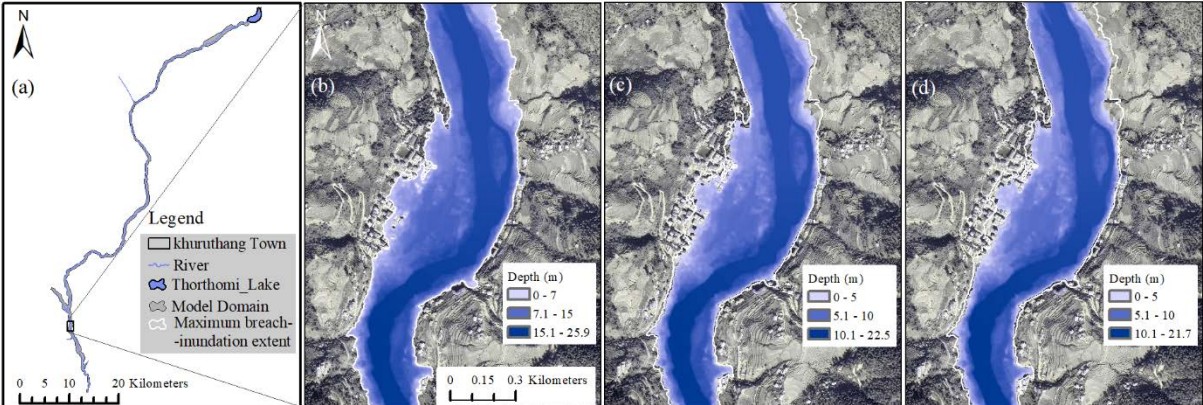

**Figure 14.** A comparison of inundation depth and extent for three breach scenarios within the Khuruthang study area. (a) A model domain highlighting Khuruthang Town. (b) The maximum breach scenario. (c) A 50% breach depth scenario. (d) A 50% breach width scenario.
Map data: Google Earth © 2023 CNES/Airbus, Maxar Technologies.

The spatial distribution of flood depth for a maximum breach scenario at different time steps at Punakha Dzong and Khuruthang town are provided in Fig. 15.





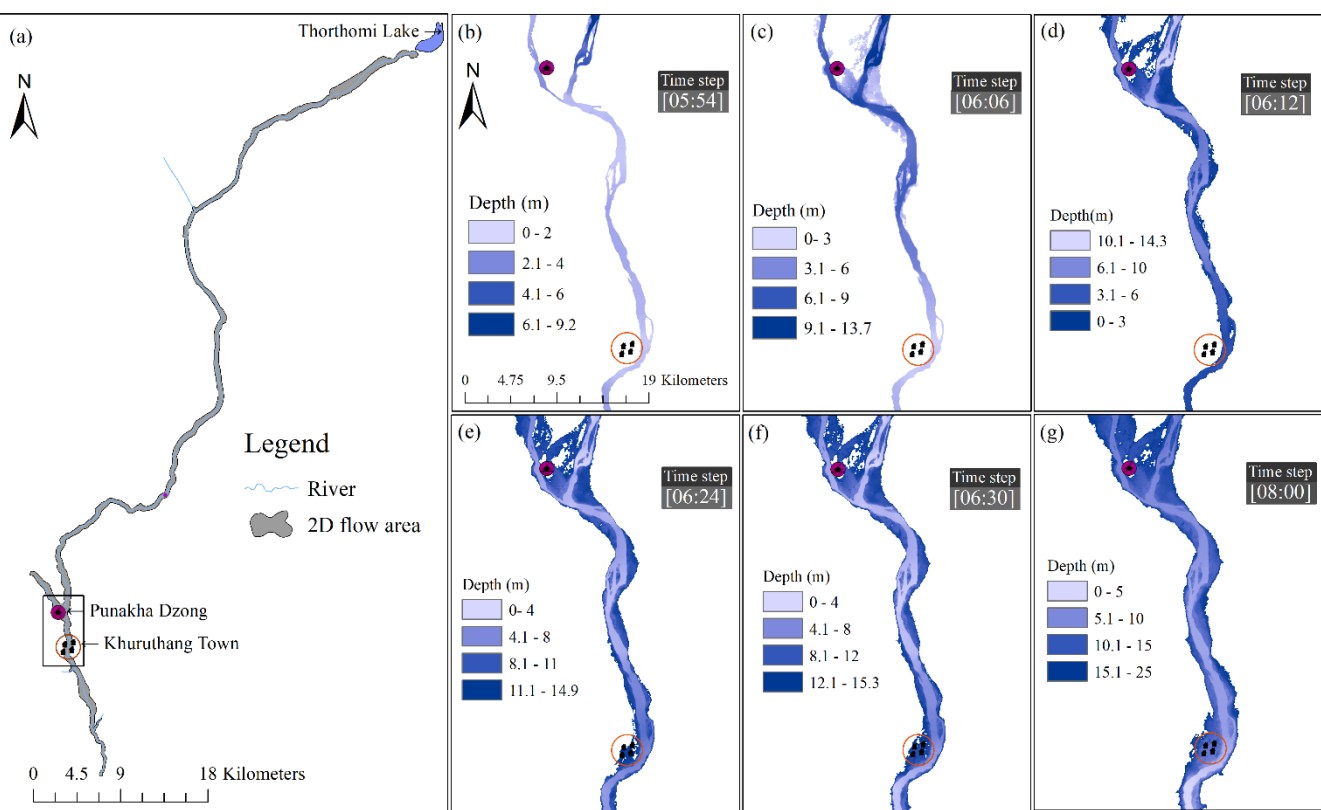

**Figure 15.** The temporal change of the spatial extent of flood depth at Punakha Dzong and Khuruthang. (a) The hydrodynamic model domain. (b - g) The inundation depth at six time steps.

Due to higher peak flow and a longer flood duration, the overall flood hazard potential for the inhabited area of the Thorthomi Lake GLOF is significantly higher, as compared to the 1994 Luggye GLOF. Most of the flood path lies in the narrow V-shaped valley, where there are few to no settlements or infrastructure. We estimate that over 1,277 houses, most in the lower region of the study area, will be inundated in a GLOF. Aside from this, infrastructures such as roads, bridges, and sand dredging equipment will be damaged.

Notable damage during the 1994 GLOF occurred in Punakha Dzong. The area near the dzong was completely inundated. As shown in Fig. 13 & 15, the simulated future GLOF indicates that the Punakha Dzong area will be completely flooded, with a maximum depth of over 10 metres.

The Punakha (the middle-downstream of the domain, see Fig. 13) and Wangdue districts (consisting of the Bajo and Jagathang settlements, and the downstream domain, see Fig. 3 & 13) are leading producers of rice, an essential crop for the country's GDP and food security. Any damage to agricultural land would have a devastating impact on farmers and the nation. Aside from potential damage to buildings and infrastructure such as roads and bridges, agricultural land would also become submerged and destroyed by a flood. We estimated that approximately 193 to 245 hectares of agricultural land will be



inundated under different scenarios in a Thorthomi GLOF event. Figure 16 shows the potential extent of floods for different land use classes and highlights probable damage to agricultural land, particularly in the areas of Samdingkha and Jagathang. The overall hazard potential of a GLOF from Thorthomi Lake under different scenarios is summarised in Table 4. Although the peak flow rate of each scenario is different (29% to 37% between the maximum and minimum for the result depicted in

Fig. 11), the total inundation area, the number of submerged buildings, and the area of impacted cultivated land are not much different (12%, 22%, and 21%, respectively), implying that the estimated flood is significant even for the most minor flood scenario (BRW-50% scenario) for the Thorthomi GLOF. The scenarios indicated that most of the damage will occur for river properties and that farmland will be substantially damaged, even when a dam breach is not drastic. The soil in farmlands will also be eroded and covered by debris. Damage to irrigation is expected and may affect agriculture in farmland located behind

flooded areas, which would make the damage to farmers' production extensive and long-term. In advance, careful evacuation planning and business continuity planning (e.g., JICA, 2015), including a plan for agriculture, are essential in order to mitigate damage caused by a future Thorthomi GLOF.

**Table 4.** The damage potential of a GLOF from Thorthomi Lake.

| Hazards→ Scenarios↓ | Total inundation area (km$^2$) | Number of buildings inundated | Total cultivated agricultural land impacted (ha) |
|---|---|---|---|
| Maximum breach | 22.7 | 1277 | 245.6 |
| ½ breach depth | 20.8 | 1044 | 206.4 |
| ½ breach width | 19.9 | 1000 | 193.4 |



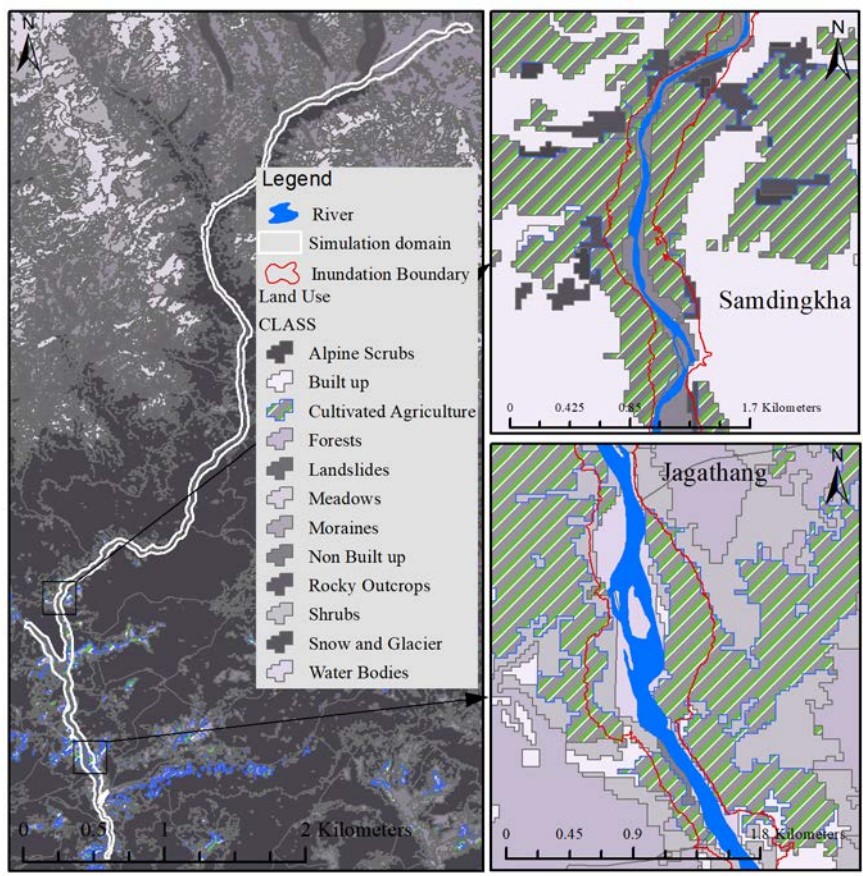

**Figure 16.** A probable GLOF inundation extent on land use classes. (Land use data source: National Land Commission Secretariat, Bhutan).

## 7 Conclusion

The future hazard and damage of a GLOF from Thorthomi Lake, one of the potentially dangerous glacial lakes in Bhutan, was explored in this study. Prior to assessing the hazards of a Thorthomi GLOF, we reconstructed the 1994 Luggye Lake GLOF in order to validate the approach used in this study and to calibrate the model. The BREACH model was used to estimate the outflow hydrograph emanating from a failure of moraine dams due to overtopping flow. Moraine materials and soil parameters used to parameterise the model were obtained from a report published by the National Center for Hydrology and Meteorology

(NCHM), Bhutan. Propagation of the GLOF was simulated using a 2D routing module in HEC-RAS for modelling unsteady flow, which is an inherent characteristic of a GLOF where there is a sharp rise in the flow hydrograph.

   The bathymetry of Thorthomi Lake was estimated based on a regression equation derived from the relationship between lake area-depth-volume found within moraine lakes. We estimated that the total volume of the lake is approximately 400 million $m^3$, with a maximum depth of 161 metres. According to the maximum breach scenario, the Thorthomi GLOF may release 283





million m³ of water in under 12 hours, with a peak flow rate of 16,360 m³/s, occurring approximately 4 hours following initiation of the breaching process. Outflow hydrographs estimated by the model were used as an input for the upstream boundary condition in hydrodynamic modelling.

Flood routing was performed in order to reach a length of approximately 115 km, and then peak discharge, the flood travel time, and flood depths at locations where settlements exist were estimated. According to the maximum breach scenario,
Punakha Dzong, which lies 90 km downstream of Thorthomi Lake and at the beginning of major settlements, would witness a peak discharge of 14,128 m³/s, approximately six hours following breach initiation. A potential GLOF from Thorthomi Lake would cause extensive agricultural and infrastructural damage for 245 hectares of agricultural lands, and, for a maximum breach scenario, 1,277 buildings are expected to be inundated. Comparable damage is also expected for two minor flood scenarios, implying that such damage is inevitable for a future Thorthomi GLOF.

The close proximity of glacial lakes in the Lunana region, especially the Thorthomi and Rapstreng Lakes (see Fig. 5) pose even greater potential risk due to the possible cascading GLOF event. The failure of lateral moraine of the Thorthomi Lake would lead to lake water breaching into Rapstreng Lake which would consequently cause the failure of its moraine dam. Such possibilities are also required to be explored in order to understand the potential risk of cascading events. Since this study considered the failure of terminal moraine in the direction of the existing outlet, it is highly unlikely for such event to occur
under the current scenario, however, no such assessment has been made in this study.

The hazard assessment of GLOF plays a crucial role for understanding and mitigating the risks associated with these devastating natural events. With this study, we conclude that the Thorthomi glacial lake poses a huge potential threat to downstream settlements and infrastructure. Such assessments enable policymakers, local communities, and relevant stakeholders to make informed decisions regarding land use planning, disaster preparedness, and early warning systems.

Since glacial environments are dynamic and subject to change due to climate variations, GLOF hazard assessments are not static. Continuous monitoring and regular reassessments of glacial lakes and associated hazards are essential when accounting for environmental shifts and ensure the effectiveness of mitigation strategies. Furthermore, there is a need for a multi-disciplinary approach in GLOF hazard assessments. Collaboration between researchers, policymakers, local communities, and other stakeholders is essential for effective decision-making, disaster preparedness, and the
implementation of mitigation measures.

Since, due to a lack of actual surveyed data, volume and maximum depth were estimated based on the statistical relationships established by past studies, a major limitation of our study is the estimated bathymetry of Thorthomi Lake. The use of actual, surveyed bathymetric data may yield a more accurate prediction, as compared to those presented. A clear water assumption for flood routing is another limitation of our study. Compared with clear water, hyper-concentrated water has different dynamic
properties, and the debris in flood water may cause substantial damage to farmland, infrastructure, and human life.



**Data availability**

This study use open and commerial data. The commertial data can be distributed following the license terms and conditions.

**Author contributions**

TW: conceptualization, data curation, method, writing original draft

RT: conceptualization, data curation, draft reviewing and editing

**Competing interests**

The authors declare that they have no conflict of interest.

**Acknowledgments**

TW acknowledges the Human Resource Development scholarship by Japan International Cooperation Agency (JICA). Japa

first author was is study use open and commarial data. The commartial data can be distributed following the license terms and conditions. Authors acknowledge Professors Yuji Toda and Takashi Tashiro and other lab. members of Hydraulic Research Laboratory, Nagoya Univesity. We also xpress our gratitude to the Dr. Shigeo Suizu, Dr. Tomoyuki Wada, and Mr. Toru Koike at Earth System Science Co., Ltd. We sincere thank to the members of the National Center for Hydrology and Meteorology for their continued support of this study.

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
