# Peer review of "A glacial lake outburst flood risk assessment for the Phochhu River Basin, Bhutan"

_Natural Hazards and Earth System Sciences, 2023_

## Author Response (AR1)

**Comments from Stefan Ram**

The authors describe a GLOF simulation by coupling a dam breach model with a hydrodynamic model. They give a good overview about their methodologies which are up to date and widely used by scientists. The input data is suitable and well described. Due to a lack of data, many assumptions had been made on the glacial lake, the main driver of the model. Those assumptions are based on widely agreed formulations, but they still don't guarantee a perfect result.

The structure of the manuscript is clear at the beginning. However, from section 6 I feel there is some inconsistency. I expect section 6 (prediction of the Thortomi GLOF) to be similar structured as section 5 (reconstruction of the 1994 Lygge GLOF). This ensures an easier comparison between both simulations. Additionally, I wouldn't put 6.5.1 and especially Table 4. In the result section as they are describing input data for the breach analysis.

(Hereafter, blue sentences provide our responses to each comment.) Thank you for your suggestions for improving structural consistency between Sections 5 and 6. Based on your suggestions, we rearranged the entire manuscript.

The conclusions are reasonable and clearly explained.

This work fills a gap of many GLOF studies in Bhutan that were already published. Its methodologies are very good and give robust results. There are some minor issued that need to be addressed before publishing this manuscript.

Minor issues:

1 – DSM

It's not clear to me what elevation data was exactly used. I guess its AW3D Standard 2.5m DSM right? A 2.2m does not exist. The 2.2m is due to reprojection of the 2.5m WGS84 original DSM to UTM. Please indicate that.

Usually DTMs (not DSM) are needed for flood modelling. FABDEM and MERIT are also some kind of DTM as buildings and vegetation are removed. I see you did some editing to the riverbed so to flow propagation should be correct. But inundation mapping where you did not touch the DSM data might lead to wrong flood extents due to hedges and especially bridges that are still represented in the elevation data.

If you say "other" freely (line 209), then people might think that AW3D 2.5m is freely available but I think it's not. Can you please clarify.

The elevation data used was AW3D Standard 2.5m resolution DSM. Resolution changes depend on location, and 2.2 m is based on the actual resolution we used in our study. For simplicity, in the revised manuscript, we refer to it as "AW3D-2.5m". Additional explanations on data resolution have been added to the revised document (Lines 369 to 376 in the revised manuscript with edit tracks).

While DTMs are required for flood modelling, most freely available data poorly represented the topography of the study area (Lines 367 to 369). A channel correction (riverbed) for the middle reach was made to enable proper flow propagation. In other areas of the studied reach, the DSM adequately represented the channel. We removed bridges and structures along the river channel. A description has been added in the revised manuscript (Lines 380-381).

2 – Figures

Please revise figure 14 and figure 15. Make sure that the class breaks if the colour ramp are equal in all tiles of the figures. With an individual colorization in each image, it is not possible to compare visually.

And the colour camp is inverted in section (d) of figure 15, please correct.

To improve clarity and correct the error, Figures 14 and 15 have been revised.

3 - Input data

I can't find the amount of inflow at Mo Chu river. In figures 13, 14 and 15 it is clearly visible that there is flow that can't be triggered by the GLOF. Especially for the interpretation of hydrographs downstream of Punaka Dzhong this is very important.

Inflow from Mochhu was not considered since the normal flow rate of the Mochhu River is small (not for a future Thortomi GLOF, but, for example, 96 m$^3$/s during the 1994 Lugge GLOF, Line 207) compared with the flow rate of a GLOF. Inundation extent and depth in the Mochhu River is due to backwater flow from the Phochhu River. This point has been added in the text (Lines 565 to 572).

4 - Language

Generally, the language is very good and easy to understand.

I have only seen the acronym PFV was introduced in line 330 but it was never used again even though it is possible.

"PFV" has been removed from the manuscript.

5 - Computation

I know HEC can simulate sub grid scale but 20m computation grid on a 2.5m raster seems to high as you lose the full power of the DSM resolution. And small structures like hedges and buildings might get lost.

We agree that using a larger computational grid size undermines the details of the topography represented in the high-resolution DSM. We tried to run a simulation using a 10 m computational grid but it took over a week to complete the simulation. To reduce computation time, a 20 m grid was used in our study.

6 - Other

Line 249: "Here, some small contributions from…"

Can you roughly indicate the contributions?

It's strange to measure a peak flow of 2,539m³/s so far downstream while the estimated peak discharge was 1,800 to 2,500 m³/s.

The contribution from the Mochhu River (96 $m^3$/s on 7$^{th}$ October 1994) has been indicated in the revised manuscript (Line 207). Since there are no gauging stations in the Dangchhu River, it is difficult to guess the contribution from this river (Lines 206 to 208). The estimation of 1,800 to 2,500 $m^3$/s was from a previous study. Our estimation was 2,455 $m^3$/s and shows better agreement with field records. (Not discussed here but we must also consider uncertainty in field records (2,539 $m^3$/s, Line 479)).

**Comments from Rayees Ahmed**

The research article entitled "A glacial lake outburst flood hazard assessment for the Phochhu River Basin, Bhutan" addresses the concerning trend of glacial lake expansion

in the Himalayan region due to glacier melting. It specifically discusses the potential risks associated with Glacial Lake Outburst Floods (GLOFs) and analyzes the potential consequences of a GLOF from the Thorthomi glacial lake in Bhutan's Phochhu River Basin using the BREACH dam breach model and HEC-RAS hydrodynamic model. The idea of the paper is appealing and well structured, however I would like authors to incorporate the following major and minor comments before it is considered for publication in NHESS.

Thank you for carefully reviewing our manuscript. We reply to each comment below.

Major Comments

In the Introduction section you have mentioned that approximately 17 million m3 of lake water was artificially released from the Thorthomi glacial lake which has lowered the water level of the lake. Did you consider this reduction of volume in your study because the volume of the water is the essential component for the GLOF simulation? How much is the current volume of water that you have used as an input to route the potential GLOF? The reduction in lake water volume due to engineering efforts may decrease the intensity of GLOF inundation parameters/flood wave (like flood extent, velocity, depth, etc).

As indicated in Table 2 (Table 3 in the revised manuscript), water volume for the Thorthomi glacial lake was estimated as 400 million $m^3$ (and described in Line 494).

Regarding water discharge from GLOF, we agree that such efforts (temporal water level lowering) are important for reducing damage caused by a GLOF. However, our understanding of the damage potential for worst-case scenarios is still limited and was the focus of our study. Due to challenging working conditions and health issues, discharging water safely is even challenging today (Lines 93-96, Lines 689-690).

Section 5.2.1. If the moraine dam is unconsolidated with loose material, then there may be a chance of piping failure as well. I suggest authors consider this mode of failure as well.

We agree that there is a potential for piping failure. However, wave overtopping is the most common trigger for dam failure (Neupane et al. 2019). Awal et al. (2010) summarized the causes of 20 GLOF events and reported that 80% of GLOF overtopping was caused by ice and rock avalanches. Begam et al. (2018) also deemed overtopping as the major failure type for GLOF events. Following these previous research studies,

for our study, we selected overtopping as the cause of a GLOF. In the revised manuscript, this point is discussed in Lines 133-134.

Furthermore, you have not mentioned how you have calculated the parameters in Table 1. I would like to see the methods of calculation of these parameters in the revised paper. This will help other researchers to calculate these parameters while working on GLOF simulations.
Section 5.2.2. How did you prepare the breach hydrograph (inflow hydrograph)? Which parameters have you considered to prepare a breach hydrograph and how did you calculate those parameters? You need to provide a detailed methodology in this section.

To estimate the breach hydrograph shown in Section 4.2.2 in the revised manuscript, we used the method described in Section 3.2. Section 3.2 includes a review of GLOF modelling (Section 3.2.1).

You did not mention the percentage of water volume released from the lake.

For the worst-case scenario considered in this study, approximately 70% of total lake water was released. To describe this scenario, changes have been made in the manuscript (Line 524).

Why have you separated the methodology and results for both lakes? You have mixed the methodology and results which makes it very difficult to understand what you are actually trying to convey. You need to be systematic and clear while revising the manuscript.

We edited the structure of the manuscript following suggestions from Reviewers and the Editor. We now believe readability has been substantially improved.

You should move all the literature to the Introduction section. In the methodology section, you need to be clear and crisp like why and how you have selected the method and how you have calculated them. I have noticed that you have started each section by referring to huge text derived from the literature.

Structure of the manuscript has been substantially edited.

The equation-derived depth and volume always provide overestimated values when you talk about GLOF simulation. Since you have mentioned bathymetry was not possible so have used empirical equations (section 6.2.2) but why have used equations proposed by Sakai (2012) to estimate the maximum depth and volume of the Thorthomi glacial lake when we have some recent integrated empirical equations developed by Qi et al., 2022 and

Thank you for introducing a great paper. When we conducted our study, this work was not available. We now mention it in Lines 507-511.

Line 375 why not piping failure?

Based on a review of previous studies (Neupane et al., 2019, Awal et al., 2010, Begam et al., 2018) and to discuss the worst-case scenario, we discussed overtopping failure. This point has been added to Lines 133-134.

Regarding bathymetry data, I suggest authors go through the GLOBathy data Khazaei et al, 2022 or other global bathymetry datasets for the comparative analysis.

Thank you for introducing a great paper. When we conducted our study, this work was unavailable. The dataset developed by Khazaei et al. (2022) does not cover the lake under consideration in this study, and the dataset is not specific to glacial lakes.

Where is uncertainty analysis? The data and mythology utilised have several associated uncertainties. How did you carry out the uncertainty analysis?

An analysis of the uncertainty determined through prediction and the confidence band has been described in Lines 495499.

You should provide a section on the limitations of the study and future research gaps.

The last part in the discussion section in the revised manuscript provides limitations and the need for further research (Lines 639-647).

Overall, the manuscript has several major and minor issues which need thorough revision.

Minor comments

Since you have envaulted the downstream impacts of the potential GLOF or you can say elements at risk therefore I suggest "Risk assessment" instead of hazard in the title.

Thank you for your comment. We followed your suggestion and edited the tile.

Merge Figures 1 and 3

We prepared Figure 1 for the Introduction and Figure 3 of the target study site. We think keeping the present composition helps separate the introductory portion of the manuscript and the specific target of our study.

Figure 2 should be more illustrative

We understand the effectiveness of illustrative explanations. Figure 2 is a simple representation of the process flowchart highlighting required models and data. We think this simplified figure makes the overall structure and content of the paper simple to understand for all types of readers.

More key findings should be added in the abstract section.

Important results from our study include: (1) selecting and composing available good methods (glacial lake bathymetry estimations, dam break models, floods and others) and data, (2) validating the composition, and (3) applying the method to a Thorthormi GLOF event. Since these are the key components of our study, we used the beginning half of the abstract to explain this information. Main findings are reflected in the Abstract.

You need to give language check throughout the text.

We attempted to describe our results in clear language and sent the manuscript for proof editing prior to submission. The journal provides English editing during the proofing phase, so we believe the language will be as clear as possible prior to publication.

Line 90 you do not need to write the abbreviation of GLOF everywhere. Mention it in the beginning and then write the short form only I,e GLOF.

Thank you for pointing out this information. We have made the necessary changes.

Line 135 include some recent studies

A recent study by Taylor et al. (2023) was cited in Line 33. Three other recent studies were referenced in Lines 133 to 134. Qi et al. (2022) was referenced in Line 508.

Line 160: Revisit the sentence it is not clear.
The sentence has been edited (Lines 317-319).

Repetition of the sentences like 330.
The connection of sentences has been improved (Lines 114-119).

Table 4 M3 should be m3
Units in the table have been corrected.

Again, in Table 2 you have not mentioned how you have calculated these parameters.
We have indicated estimated parameter sources below the table. Most of the parameters were obtained from published papers and reports. Parameters estimated by our study were obtained from the terrain model and the regression analysis discussed in the paper.

Fig 9 nothing is clear.

The journal suggests that figures be clear for readers with color vision deficiencies. Colors and textures were chosen following this suggestion.

Fig 16. The color scheme for Lulc is confusing.

The journal suggests that figures be clear for readers with color vision deficiencies. Colors and textures were chosen following this suggestion. The figure has been revised to improve visibility.

The authors should check the grammar and language throughout the text.

We attempted to describe our results in clear language and sent the manuscript for proof editing prior to submission. The journal provides English editing during the proofing phase, so we believe the language will be as clear as possible prior to publication.

**Comments from Pascal Haegeli, Editor**

Dear authors:

Thank you very much for submitting your responses to the reviewer comments. I apologize that it has taken me so long to get back to you.
I am happy with most of your answers and look forward to the revised version of your manuscript. However, I have a few additional comments that I would like you to consider when revising your manuscript.

1) Manuscript structure
Despite both reviewers commenting on challenges related to the non-traditional structure of your manuscript, your responses do not seem to indicate that you are considering revising the structure of your paper. I would like to reiterate the reviewers' comments that a more traditional structure (introduction, background, methods, results, discussion, conclusion) would likely make your manuscript easier to read and therefore more accessible. I disagree with your statement that your study consists of three different components that need to be described in separate methods and results sections. They all contribute to the overall objective of the study and can therefore be described meaningfully in the traditional structure. While writing it this way might take some extra work for you, it will make it much easier for readers as they generally expect this structure. Hence, please take the recommendations of the reviewers to heart and explore whether a different writing structure improves the readability of the manuscript. Having a dedicated discussion section will also allow you to properly summarize and highlight the overall insights and implications from your study, which is currently missing, or at least not where readers expect it. In addition, your limitation section should go at the end of this discussion section. It is too late to discuss this in the conclusion section.

The structure of the manuscript has been substantially edited to follow a traditional structure: introduction, materials and methods, results, discussion, and conclusion. The discussion section has been reinforced. Limitations of the study have been moved to the end of the discussion section.

2) Hazard versus risk in title
I encourage you to clearly distinguish between the terms risk and hazard (see your response to the second reviewer's comment on your title). If your study includes a risk assessment, which I think it does, the title should reflect this even if you had to make certain simplifications in your analysis. I disagree with your statement that "hazard can also include risk (which, of course, depends on definition)" since these two terms should describe distinct

concepts. Using these terms imprecisely and somewhat interchangeably contributes to the misuse or inaccurate use of the terms and concepts.

The title has been revised.

3) Typo in response

Finally, I assume that in your response to the second reviewer's comment on the failure mode, you intended to say that "wave overtopping is the [most?] common (not is not the common) trigger for dam failure.

Sure, it was our typo and has been corrected.

Lastly, we thank you for fair and constructive comments, which helped improve the manuscript.

---

## Author Response (AR2)

Dear Dr. Pascal Haegeli:

Thank you very much for handling our manuscript and giving us careful suggestions to improve its quality. Below, we would like to explain our reply to each comment with blue-colored text.

Best regards,
Ryota Tsubaki, as a representative of the authors.

**Comments from Pascal Haegeli**

Dear Drs. Wangchuk and Tsubaki:

Thank you very much for revising your manuscript "A glacial lake outburst flood risk assessment for the Phochhu River Basin, Bhutan" and resubmitting it to NHESS. As pointed out by the reviewers, your study addresses an important knowledge gap. I have now examined your revisions and studied the updated manuscript in detail. I believe that your edits improved the quality of the manuscript tremendously. However, I have a series of additional suggestions that aim to further improve the clarity and accessibility of your manuscript. Please see below for details. I would appreciate it if you could address these comments before I accept your manuscript for publication.

I hope you find my comments useful and they help you bring your manuscript to the next levels. Please let me know if you have any questions or require additional information.

Pascal Haegeli, PhD

NHESS Editor

Simon Fraser University, Vancouver BC, Canada

Introduction

L41 – Fig. 1: Please move Fig. 1 to the end of Section 1.3 after you introduced the figure in the text for the first time (L60). Figures should always follow their introduction.

Thank you very much for the suggestion. We moved the location of the figure.

L41 – Fig. 1: Adding major cities and transportation routes to the map might make it easier for readers to understand the map and put other map features in context.

Thank you for the suggestion. We edited the figure as suggested.

L41 – Fig. 1: Please introduce the PDGL abbreviation in the caption of the figure. Maybe the term could be written out in the legend.

We edited the figure caption.

L41 – Fig. 1: It might be useful to highlight Luggye Lake, Tshojo Lake, and Thorthomi Lake in Figure 1 to allow readers to see where these lakes are.

We edited the figure as suggested.

L61: Does "the largest in the country" relate to the size of the river basin? If that is the case, reword the sentence to "The Punatsangchhu River Basin (the largest in the country) contains eleven PDGLs".

The Punatsangchhu River Basin contains the largest number of PDGLs per basin in the country. This point is explained in the revised manuscript.

L80+ – Section 1.5: This section seems at the wrong spot as it does not seem to justify the study objective and disrupts the flow of Sections 1.4. and 1.6. Instead, it highlights a limitation of the existing data/understanding, which has implications on your modelling approach. Hence, it seems more meaningful to include this information in the methods section.

We described the lack of direct data for GLOF bathymetry here because one of the main objectives of our study was to estimate the bathymetry of the focused GLOF. However, we understand that the flow is not smooth, so we moved the information to Section 3.1.

L94: Start this section with "To contribute to these risk management efforts, ..." (or similar) to better tie this paragraph to the research need outlined in Section 1.4.

Thank you for the suggestion to improve the connection between sub-sections. Based on the suggestion, we edited the section.

L94+ - Section 1.6: You should use present tense to describe what your study does in the introductory section. This is different from the methods section.

We corrected the tense in the sub-section.

L100: The sentence "Since the overtopping …" seems out of place. You could either move it up to L94 where you introduce the objective of the study or to the methods section.

The sentence has been moved.

L110+ – Section 1.7: Please write this section a little bit more interesting and not just have a sequence of similar sentences: "Section 2 describes… Section 3 describes … Section 4 reports … Section 5 discusses …"

We attempted to put some flavor in the description of paper structure.

L115 – Figure 2: This figure seems too detailed and out of place for the introduction section. It would be better to move it to the beginning of the methods section (see later comment).

We moved this figure to the methods section.

Study area and GLOF event

L121: The description of the Punatsangchhu River Basin and its link to Fig. 3 is a bit confusing as the figure seems to primarily focus on the Phochhu sub-basin. Please clarify.

We edited the caption and the text to clarify the location of the Punatsangchhu River Basin.

L121: It would make the section easier to read if the physical characteristics (location, area, discharge volumes, …) were discussed together and the population and administrative characteristics explained together in a separate paragraph. They are mixed right now.

We edited the section based on the suggestion.

L130 – Figure 3: Note that the outline of the study area/Phochhu sub-basin (I believe) shown in Fig. 3 is different from what is shown in Fig. 1. In my opinion, Fig. 3 could be substantially simplified by a) eliminating panel (b) and already outlining the study area in Fig. 1, and b) combining panels (a) and (c) into a single annotated map. Please also note that the labels of the different panels (a, b, c) do not line up with the labels in the caption and the text!

Fig. 3 (Fig. 2 in the revised manuscript) shows the study area, which is not identical to the Phocchu Basin (the study area is the catchment area of the PHPA-I & II excluding the Mocchu River Basin). For clarity, we kept panel (b) (panel (a) in the revised manuscript) and edited the figure caption.

L144+: The paragraph starting at this line does not seem to fit under the heading "2.2 Thorthomi glacial lake" as it describes the social and economic characteristics of the downstream areas. Please consider combining Sections 2.1, 2.2., and 2.3 into a single section that describes the study area. There should be separate paragraphs describing the physical and economic characteristics of the region.

We edited the text based on the suggestion.

L155: It is not considered good style to write "As shown in Figure 4, ...". Instead write "The lake is one of four glacial lakes in an area that spans a few kilometres (Figure 4) and had an outburst in 1994." This is much more concise. Also, it is not necessary to write "(see Figure 4)" (e.g., L161). Just write "(Figure 4)".

We edited the text based on the suggestion (we changed "(see Figure X)"s to "(Figure X)"s in other places too).

L177: The sentence "We constructed ..." seems out of place because it describes the analysis approach and not just the event as implied by the section heading on L154.

This sentence has been move to the beginning of Section 3 and edited to fit the description there.

Materials and methods

L179 – Materials and methods: I recommend you start this section with a brief overview of your analysis approach that includes Fig. 2. This would provide valuable overview and context for the content presented in the following subsections.

The figure was edited and an explanation has been added to the beginning of Section 3.

L180+ – Section 3.1: The heading for this section is misleading as you are only presenting an approach for estimating lake volumes and not a regression analysis. Please consider

using a more appropriate title like "Estimating lake volumes". The section also goes back and forth multiple times between the mean depth (e.g., Huggel et al, 2002; Cook and Quincey, 2015) and the maximum depth approaches (Sakai, 2012). For example, on L195, you mention that the approach of Cook and Quincey (2015) included the dataset of Huggel et al (2002), which was already mentioned on L195. Please improve the structure of this section to make it less convoluted for the reader.

The sub-section title has been changed to "Estimating geometries of glacial lakes". The paragraphs have been edited to improve readability.

L218: Please explain why you chose to use the approach of Sakai (2012).

For predicting moraine dam breach processes, the maximum depth is crucial, so we employed the equations proposed by Sakai (2012).

L222 – Section 3.2: Here is the place to let the reader know that you are focusing on overtopping of lake water as the failure mode. See earlier comment.

We edited the text based on the suggestion.

L241: The sentence "Unlike parametric models, physically based breach models consider the geotechnical aspects of dam materials, as well as …" should go earlier when you introduce the BREACH model for the first time. This section should focus on what you did with the model and why.

We edited the text based on the suggestion. We added the reason why we used BREACH: "because of its better predictive accuracy for future extraordinal GLOF events".

L271: Can you elaborate on how the data was adopted from available reports and research documents.

The dam breach model requires detailed data regarding moraine dam materials. No single source or paper contains the parameter data required. As a result, we used some data from Koike and Takenaka (2012) and some data from the publication of NCHM (2019a).

L271: The sentence "Table 2 provides …" is unnecessary. Just cite the table in the previous sentence. Similar to earlier comment.

We edited the text based on the suggestion.

L274 Table 1 & 2: These two tables could potentially be combined with the parameter names in the first column and the parameter estimates for the Luggye and Thorthomi Lakes in columns 2 and 3 respectively. This would allow readers to compare the parameters estimate between the two lakes more easily.

We agree that it is good idea and edited the text based on the suggestion.

L 281 & 289: It is only necessary to introduce the HEC-RAS abbreviation once.

We introduce the abbreviation of HEC-RAS in Section 1.

L 305: Please avoid subheadings at a fourth level (e.g., 3.3.2.1). The information in this section can easily be presented in just two paragraphs without subheadings.

Fourth level subheadings have been removed in the revised manuscript.

L316: It might be useful to state here that three different DEMs are available for the study area. This would set the stage for the descriptions of the different DEMs that follow.

The manuscript has been revised based on the suggestion.

L329: The sentence "Figure 6 illustrates ..." is not necessary. Just cite the figure in the previous sentence. Similar to earlier comments.

We edited the text based on the suggestion.

L357 – Section 3.3.3 and 3.3.4: These two sections can probably be combined into a single section that explains the hydrodynamic modelling at the two study sites. This would result in the following streamlined subheadings for the flood routing section: 3.3.1 Hydrodynamic model; 3.3.2 Ground elevation data; and 3.3.3 Implementation for GLOF reconstruction and prediction.

The structure was edited based on the suggestion.

L367: Delete "almost that was" so that the sentence reads "We used a hydrodynamic model similar to ...".

We edited the text based on the suggestion.

Results

L375 (and other sections): Try not to start sentences with "As shown in Figure X. Instead, just cite the figure at an appropriate location in the first sentence. Similar to earlier comments.

We edited the text based on the suggestion.

L407 – Table 3: Is it necessary to include the measured and estimated values of all these PDGL in this manuscript? It is unclear to me who this relates to the information presented in this study.

Since it added little or no major contribution to the manuscript, we removed the Table.

L443+: The information presented in this paragraph seems to belong to the methods section.

We edited the text based on the suggestion.

Discussion

L455: Simplify the heading to just "Discussion"

We edited the text based on the suggestion.

L474 – Figure 13: Please reverse panels d) and e) to produce the regular top-to-bottom and left-to-right flow.

We modified the layout of the figure.

L474 – Figure 13-15: If possible, if would increase the font size of the labels in these figures. The time step labels in Fig. 15 are particularly difficult to read.

We enlarged the labels in the figures and added labels (a) to (g) in Figure 15.

Conclusion

No suggestions or comments.

General

Language: The writing of your manuscript is still not at the level it should be for publication. The paragraph and sentence structures are often convoluted, there are way too many commas in the writing. While you are correct that Copernicus staff will copy-edit the manuscript before sending it into production, their role is to fix error and not the make the writing easier to understand. I therefore recommend that you work through the text of the manuscript in detail again.

We reviewed the whole text in the manuscript and tried to improve its readability.

Finally, we would like to express our sincere thanks to the review and editorial teams for giving us careful comments and suggestions to improve the quality of the manuscript.

---

## Author Response (AR3)

Dear the editorial team of NHESS:

Here, we uploaded manuscript PDF, Word file, and figure files. When preparing the separate figure files, we saved the figure files as EPS format, which is recommended on the submission page. During this preparation, some small texts in figures are enlarged to improve readability. The layout of Figure 6 had also switched following the referring in the main text (panel c was referred to first, then panels b and a were referred to, but after this layout change, panels a, b, and c are referred to in this order.) All the above changes do not change scientific content, but we are trying to improve the manuscript's readability, especially for those with vision deficiencies.

Best regards,
Ryota Tsubaki